# Indian genetic heritage in Southeast Asian populations

**Piya Changmai**[1]*, **Kitipong Jaisamut**[1], **Jatupol Kampuansai**[2,3], **Wibhu Kutanan**[4], **N. Ezgi Altınışık**[1¤], **Olga Flegontova**[1], **Angkhana Inta**[2,3], **Eren Yüncü**[1], **Worrawit Boonthai**[5], **Horolma Pamjav**[6], **David Reich**[7,8,9,10], **Pavel Flegontov**[1,7,11]*

1 Department of Biology and Ecology, Faculty of Science, University of Ostrava, Ostrava, Czech Republic,
2 Department of Biology, Faculty of Science, Chiang Mai University, Chiang Mai, Thailand, 3 Research Center in Bioresources for Agriculture, Industry and Medicine, Chiang Mai University, Chiang Mai, Thailand, 4 Department of Biology, Faculty of Science, Khon Kaen University, Khon Kaen, Thailand, 5 Research Unit in Physical Anthropology and Health Science, Thammasat University, Pathum thani, Thailand, 6 Hungarian Institute for Forensic Sciences, Institute of Forensic Genetics, Budapest, Hungary, 7 Department of Human Evolutionary Biology, Harvard University, Cambridge, Massachusetts, United States of America, 8 Department of Genetics, Harvard Medical School, Boston, Massachusetts, United States of America, 9 Broad Institute of MIT and Harvard, Cambridge, Massachusetts, United States of America, 10 Howard Hughes Medical Institute, Harvard Medical School, Boston, Massachusetts, United States of America, 11 Kalmyk Research Center of the Russian Academy of Sciences, Elista, Kalmykia, Russia

¤ Current address: Human-G Laboratory, Department of Anthropology, Hacettepe University, Ankara, Turkey

* piya.changmai@osu.cz (PC); pavel.flegontov@osu.cz (PF)

**Data Availability Statement:** All genotype data of 119 individuals in this study is publicly available at the Reich lab website (https://reich.hms.harvard.edu/datasets).

## Abstract

The great ethnolinguistic diversity found today in mainland Southeast Asia (MSEA) reflects multiple migration waves of people in the past. Maritime trading between MSEA and India was established at the latest 300 BCE, and the formation of early states in Southeast Asia during the first millennium CE was strongly influenced by Indian culture, a cultural influence that is still prominent today. Several ancient Indian-influenced states were located in present-day Thailand, and various populations in the country are likely to be descendants of people from those states. To systematically explore Indian genetic heritage in MSEA populations, we generated genome-wide SNP data (using the Affymetrix Human Origins array) for 119 present-day individuals belonging to 10 ethnic groups from Thailand and co-analyzed them with published data using PCA, ADMIXTURE, and methods relying on $f$-statistics and on autosomal haplotypes. We found low levels of South Asian admixture in various MSEA populations for whom there is evidence of historical connections with the ancient Indian-influenced states but failed to find this genetic component in present-day hunter-gatherer groups and relatively isolated groups from the highlands of Northern Thailand. The results suggest that migration of Indian populations to MSEA may have been responsible for the spread of Indian culture in the region. Our results also support close genetic affinity between Kra-Dai-speaking (also known as Tai-Kadai) and Austronesian-speaking populations, which fits a linguistic hypothesis suggesting cladality of the two language families.

**Funding:** This work was supported by the Czech Ministry of Education, Youth and Sports: 1) Inter-Excellence program, project #LTAUSA18153; 2) Large Infrastructures for Research, Experimental Development and Innovations project "IT4Innovations National Supercomputing Center – LM2015070". E.Y., O.F., P.C., and P.F., were also supported by the Institutional Development Program of the University of Ostrava (IRP201825). J.K. and A.I. acknowledge partial support provided by Chiang Mai University, Thailand. W.K. was supported by Khon Kaen University. P.F. was also supported by a subsidy from the Russian federal budget (project No. 075-15-2019-1879 "From paleogenetics to cultural anthropology: a comprehensive interdisciplinary study of the traditions of the peoples of transboundary regions: migration, intercultural interaction and worldview"). D.R. was supported by the National Institutes of Health (NIGMS GM100233), the John Templeton Foundation (grant 61220), and by the Allen Discovery Center program, a Paul G. Allen Frontiers Group advised program of the Paul G. Allen Family Foundation; D.R. is also an Investigator of the Howard Hughes Medical Institute. The funders had no role in study design, data collection and analysis, decision to publish, or preparation of the manuscript.

**Competing interests:** The authors have declared that no competing interests exist.

## Author summary

Mainland Southeast Asia is a region with great ethnolinguistic diversity. We studied genetic population history of present-day mainland Southeast Asian populations using genome-wide SNP data. We generated new data for ten present-day ethnic groups from Thailand, which we further combined with published data from mainland and island Southeast Asians and worldwide populations. We revealed South Asian genetic admixture in various mainland Southeast Asian ethnic groups which are influenced by Indian culture but failed to find it in groups that remained culturally isolated until recently. Our finding suggests that migrations of Indian people in the past may have been responsible for the spread of Indian culture in mainland Southeast Asia. We also found support for a close genetic affinity between Kra-Dai- and Austronesian-speaking populations, which fits a linguistic hypothesis suggesting cladality of the two language families.

## Introduction

Mainland Southeast Asia (MSEA) is a region with high ethnolinguistic diversity and complex population history. Hundreds of indigenous languages belonging to five major language families (Austroasiatic, Austronesian, Hmong-Mien, Kra-Dai, and Sino-Tibetan) are spoken in MSEA [1]. Archaeological evidence shows that anatomically modern humans migrated to MSEA roughly 50000 years ago [2,3]. Previous archaeogenetic studies indicate that the earliest MSEA individuals belong to the deeply diverged East Eurasian hunter-gatherers [4]. Andamanese hunter-gatherers (Onge and Jarawa) and MSEA Negritos are present-day populations with substantial proportions of ancestry from the deeply diverged East Eurasian hunter-gatherer lineage [4,5]. Neolithic populations in MSEA were established by admixture between these local hunter-gatherers and agriculturalists who migrated from South China around 4000 years ago [4,5]. The genetic makeup of MSEA Neolithic individuals is similar to present-day Austroasiatic-speaking populations [4,5]. That pair of studies also detected additional waves of migrations from South China to MSEA during the Bronze and Iron Ages. There is evidence of trading in Indian goods in MSEA and of glass bead manufacturing by MSEA locals using Indian techniques during the Iron Age [2]. Early states in MSEA during the first millennium CE, such as the Pyu city-states, Funan, Dvaravati, Langkasuka, and Champa were established with a substantial influence from Indian culture [6]. A Chinese source described Funan, one of the earliest known states in MSEA, as established by an Indian Brahmin named Kaudinya and a local princess [2,6]. The spread of Indian culture had various impacts on the region, such as state formation, laws, religions, arts, and literature. Ancient Sanskrit inscriptions were found throughout MSEA, and several present-day languages in the region contain numerous Sanskrit loanwords [6].

Previous studies based on uniparental markers found West Eurasian-associated haplogroups in some populations in MSEA, which is possibly a signal of South Asian admixture [7–10]. Some genome-wide studies previously documented South Asian admixture in few MSEA populations [11–13], but many ethnic groups in the region remain unexplored, especially Austroasiatic-speaking populations from Thailand which have strong Indian cultural connections. Some previous studies of MSEA populations did not focus on South Asian influence in the region, and South Asian ancestry was sometimes overlooked in those studies [14–16].

Thailand is a country in the middle of MSEA, and many ancient Indianized states were located in its territory [6]. Various present-day populations are possibly descendants of people

from ancient Indianized states in the region, as they have inherited languages and culture from those states, and they reside in regions that once were within the territory of the ancient states [6]. In Thailand, 51 indigenous languages from five major language families are spoken [1]. The official language of Thailand is Thai, a language of the Kra-Dai language family [1]. A previous linguistic study proposes that the expansion of Southwestern Tai (a branch of the Kra-Dai language family) from present-day China into MSEA began between the 8th and 10th centuries CE [17]. A linguistic connection between Kra-Dai and Austronesian language families was previously proposed [18]. Sagart proposed that Austronesian-related Kra-Dai ancestors migrated from Taiwan to settle around Guangdong coast around 4,000 YBP [19]. Interaction with local people resulted in relexification of Kra-Dai languages [19,20]. A study on uniparental markers also supports genetic connection between Austronesian and Kra-Dai-speaking populations [8].

We generated genome-wide SNP genotyping data for ten populations from Thailand: six Austroasiatic-speaking populations (Khmer, Kuy, Lawa, Maniq, Mon, and Nyahkur), one Hmong-Mien-speaking population (Hmong), one Kra-Dai-speaking population (Tai Lue), and two Sino-Tibetan-speaking populations (Akha and Sgaw Karen). Akha, Lawa, Karen, and Hmong are officially recognized as hill tribes (a term commonly used in Thailand for minority ethnic groups residing mainly in the northern and western highland regions of the country) in Thailand [21]. Another group genotyped in this study, Khmer from Thailand, is a Northern Khmer-speaking population which is closely related to Cambodian Khmer (Cambodians), the majority population in Cambodia [1]. Present-day Khmer are likely to be descendants of people from ancient Khmer states in the region [22]. Kuy is a population which has interacted with the Khmer since ancient times [23]. The Mon and Nyahkur languages belong to the Monic branch of the Austroasiatic family [1]. These populations are probably related to people from ancient Mon states in present-day Thailand and Myanmar [22]. Tai Lue is a Kra-Dai speaking group, which is closely related to Dai from Southern China [1]. Maniq, a MSEA Negrito group, are present-day hunter-gatherers. We combined our data with published MSEA and worldwide data. The aims of our study are: 1) to explore South Asian admixture in MSEA populations to find out if the Indian cultural expansion was driven by cultural diffusion or movement of people from India and subsequent inter-marriage with local MSEA people; 2) to investigate population structure in MSEA groups; and 3) to study the genetic connections between Kra-Dai and Austronesian-speaking populations since a sister-clade relationship between the two languages families was previously suggested.

## Results

### Overview of the genetic makeup of East and Southeast Asian (ESEA) populations

Using the Affymetrix Human Origins SNP array [24], we generated genome-wide genotyping data (574,131 autosomal sites after quality control filtering) for 10 present-day human populations from Thailand (Fig 1). We merged our data with published data for ancient and present-day worldwide populations (S1 Table). To obtain an overview of population structure, we performed principal component analysis (PCA) (Fig 2). South Asian (SAS) populations lie on a previously described North-South cline [25]. Central Asian and Siberian populations lie between the European (EUR)—SAS cline and the East and Southeast Asian (ESEA) cluster. In agreement with expectations from geography, the Central Asian cline lies between the Siberian and South Asian clines. Maniq, MSEA Negritos (NEGM), are located between the ESEA cluster and Onge, the Andamanese Negritos (NEGA). Munda populations, Austroasiatic-speaking populations from India which were shown in a previous study [26] to be a genetic mixture of

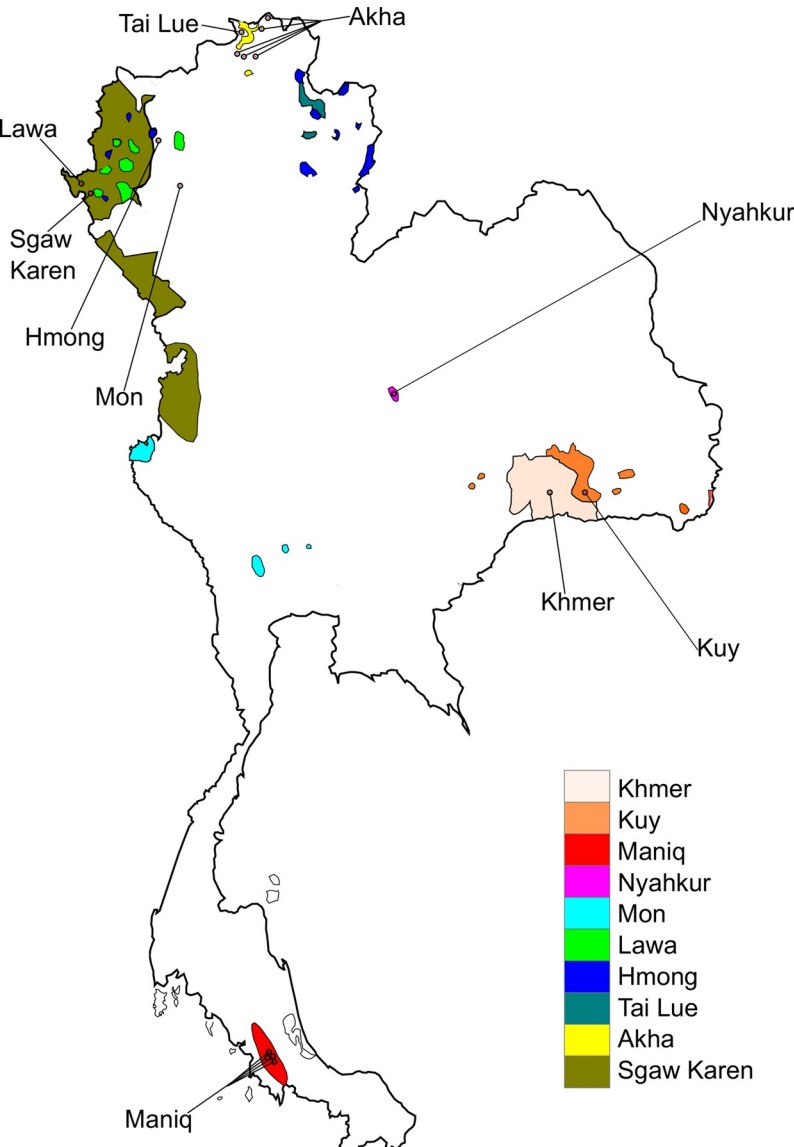

**Fig 1. Locations of populations for whom genome-wide data was generated in this study.** The colored areas on the map represent the geographic distribution of their languages (adapted from Eberhard 2020 [1]). Black dots at the end of the lines indicate the sampling locations. These populations speak languages from four families: Austroasiatic (Khmer, Kuy, Lawa, Maniq, Mon, and Nyahkur), Hmong-Mien (Hmong), Kra-Dai (Tai Lue), and Sino-Tibetan (Akha, Sgaw Karen). The map was plotted using an R package "rnaturalearth" (https://github.com/ropensci/rnaturalearth) with Natural Earth map data (https://www.naturalearthdata.com/).

South Asian and Southeast Asian populations, lie between the SAS cline and ESEA cluster, as expected (Fig 2). Populations from East and Southeast Asia form a well-defined cluster, but the positions of some populations such as Sherpa, Burmese, Mon, Thai, Cambodian Khmer (Khmer C), Cham, Ede, Malay, Khmer from Thailand (Khmer T), Nyahkur, and Kuy are shifted towards the SAS cline (Fig 2).

Next, we performed a model-based clustering analysis using the ADMIXTURE approach. At 12 hypothetical ancestral populations, Mon, Khmer from Thailand, Kuy, Nyahkur, Burmese, Thai, Cambodian Khmer, Cham, Ede, Giarai, and Malay (data for four former populations were generated in this study) demonstrated a "light pink" ancestry component

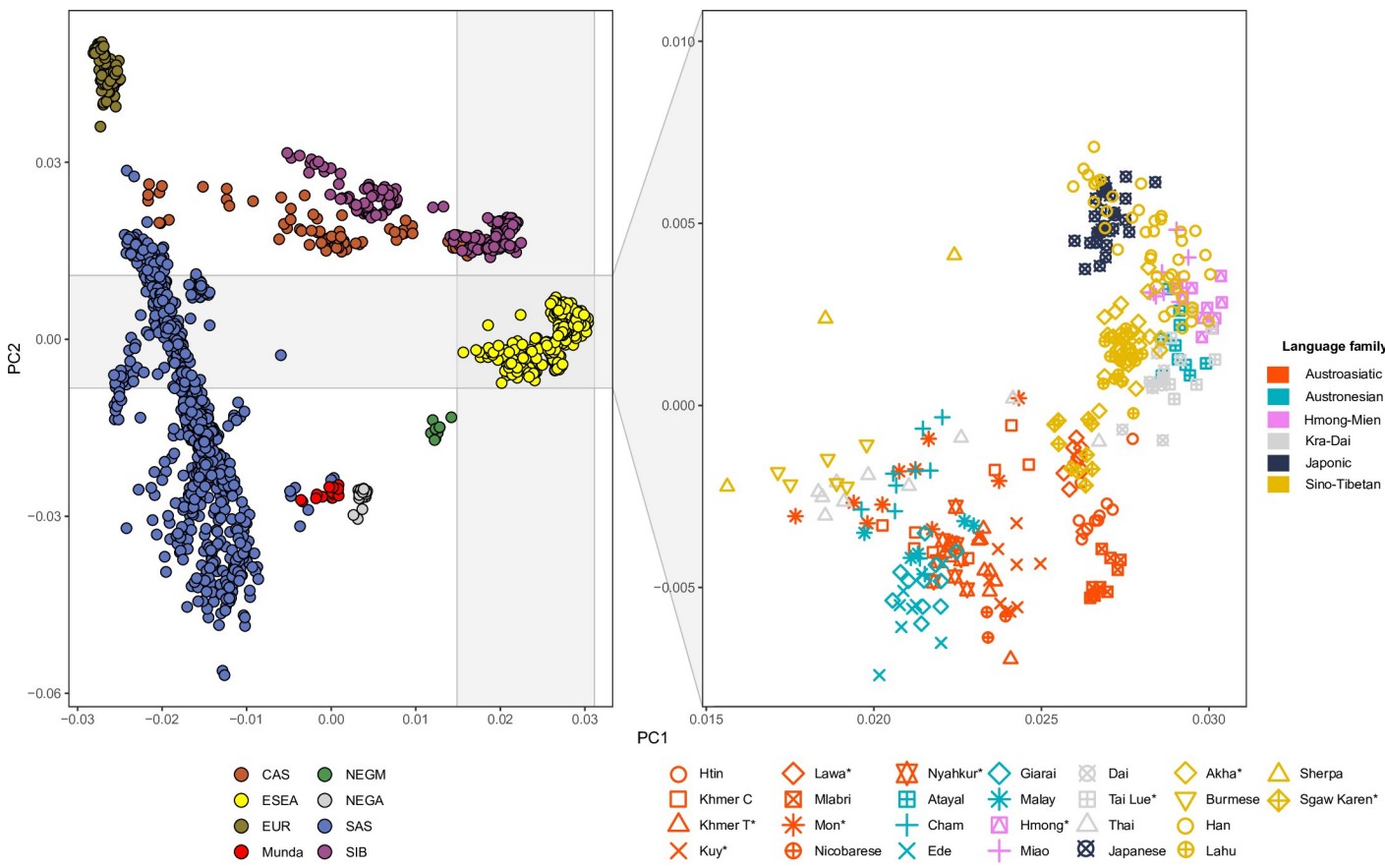

**Fig 2. A principal component analysis (PCA) plot of present-day Eurasian populations. Left panel**: An overview of the PC1 vs. PC2 space for all populations. The legend at the bottom of the plot lists abbreviations of meta-populations: CAS, Central Asians; ESEA, East and Southeast Asians; EUR, Europeans; Munda, Austroasiatic-speaking populations (the Munda branch) from India; NEGA, Andamanese Negritos; NEGM, Mainland Negritos; SAS, South Asians; and SIB, Siberians. **Right panel**: A zoomed-in view on the rectangle in the left panel. Colors of the markers represent language families. Asterisks after population names indicate that these populations are newly genotyped in this study.

(accounting for more than 5% of their ancestry, on average) that is enriched in South Asian populations such as Irula and Mala from Southern India (Fig 3).

Outgroup $f_3$-statistics are used for measuring shared genetic drift between a pair of test populations relative to an outgroup population. We further explored hypothetical SAS admixture in MSEA by inspecting a biplot of outgroup $f_3$-tests (Figs 4 and S1). In the coordinates formed by statistics $f_3$(Mbuti; Han, an ESEA group) and $f_3$(Mbuti; Brahmin Tiwari, an ESEA group) (Fig 4), most ESEA populations demonstrate a linear relationship between the genetic drift shared with Han and the drift shared with Brahmin Tiwari. However, the positions of Burmese, Mon, Cham, Nyahkur, Khmer from Cambodia and Thailand, Kuy, Malay, Nicobarese, Giarai, and Ede are shifted from that main ESEA trend line. This shift can be interpreted as an elevated shared drift between the SAS group and the test population, as compared to other ESEA populations. Similar results were generated when we replaced Han and Brahmin Tiwari with Dai and Coorghi, respectively (S1 Fig).

### Fitting admixture models using *qpWave*, *qpAdm*, and *qpGraph*

Previous studies indicate that deeply diverged East Eurasian hunter-gatherers (associated with the Hoabinhian archaeological culture), which are related to present-day Andamanese hunter-

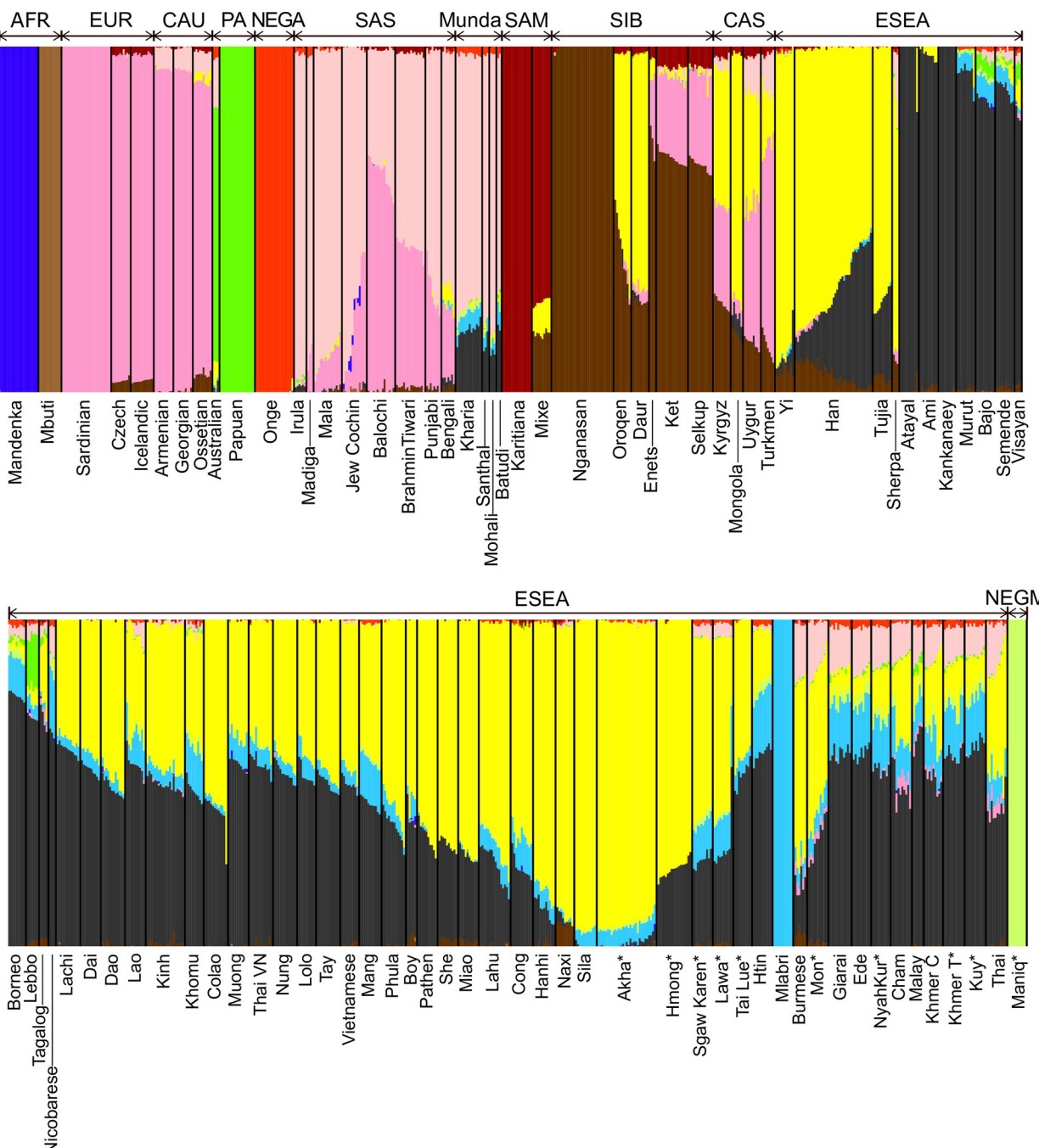

**Fig 3. Results of an ADMIXTURE analysis.** The plot represents results for 12 hypothetical ancestral populations. Abbreviations of meta-populations are shown above the plot: AFR, Africans; EUR, Europeans; CAU, Caucasians; PA, Papuans and Australians; NEGA, Andamanese Negritos; SAS, South Asians; Munda, Austroasiatic-speaking populations (the Munda branch) from India; SAM, Native South Americans; SIB, Siberians; CAS, Central Asians; ESEA, East and Southeast Asians; and NEGM, Mainland Negritos. Asterisks after population names indicate that these populations are newly genotyped in this study.

gatherers (e.g., Onge), were the first known anatomically modern humans who occupied MSEA [4,5]. MSEA populations in the Neolithic period can be modelled as a mixture of local Hoabinhians and populations who migrated from East Asia [4,5]. Our PCA and ADMIX-TURE results were not informative about Hoabinhian-related ancestry. For instance, the

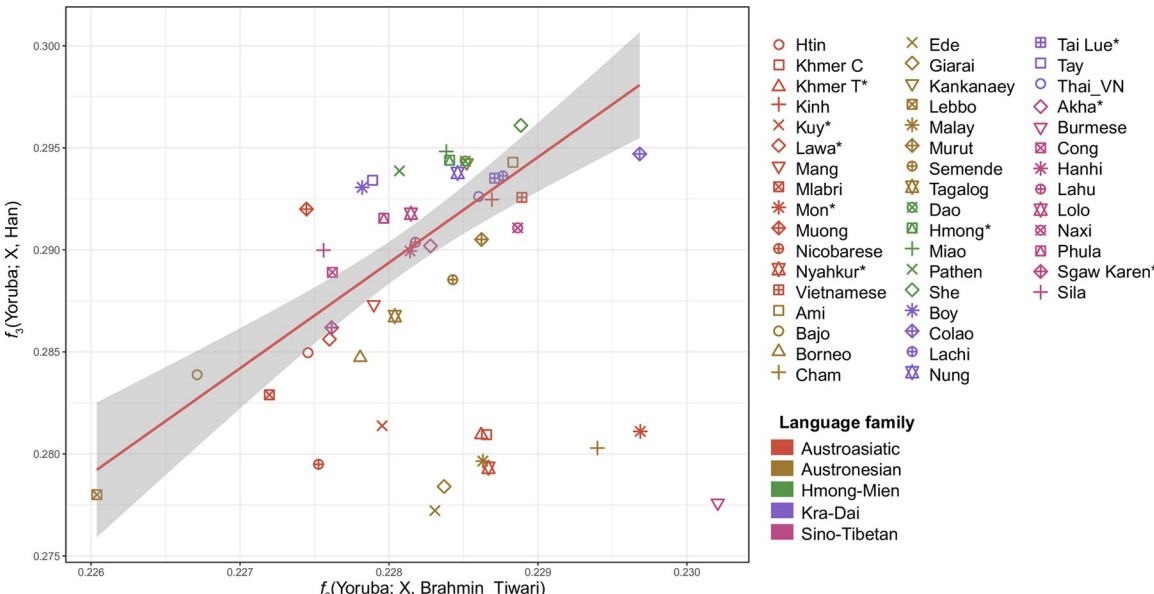

**Fig 4. A biplot showing results of outgroup $f_3$-tests.** The biplot of $f_3$(Mbuti; Brahmin Tiwari, X) vs. $f_3$(Mbuti; Han, X) illustrates the amount of genetic drift shared between ESEA populations and Brahmin Tiwari or Han. The trend line represents a ratio of shared genetic drifts that is common for most ESEA populations. The positions of few ESEA populations deviate from the trend line, which indicates elevated shared drift between the Indian reference population and the test population, as compared to most ESEA populations. Colors of the markers represent language families. Asterisks after population names indicate that these populations are newly genotyped in this study.

Maniq and Mlabri groups probably underwent population size bottlenecks, and ADMIXTURE models their genomes as derived almost entirely from a single ancestral component not detected in other populations. The problem is that genetic drift in these groups obscures their previous history when analyzed with ADMIXTURE [27].

To overcome this problem, we tested admixture models using the *qpWave* [28] and *qpAdm* methods [29,30]. We used both methods to test the plausibility of a proposed admixture model given a set of outgroups (reference populations, which are differentially related to test populations). We also used *qpAdm* to infer admixture proportions of surrogates in a target population. We used Atayal, Dai, and Lahu as alternative ESEA surrogates. These populations speak languages which belong to three different families: Austronesian, Kra-Dai, and Sino-Tibetan (the Tibeto-Burman branch), respectively. Onge, a representative of Andamanese Negritos (NEGA), was used as a surrogate for the deeply diverged East Eurasian hunter-gatherers. Fifty-five populations composed of at least five individuals were used as South Asian surrogates (S2 Table). Outgroups ("right populations") for all *qpWave* and *qpAdm* analyses were the following diverse present-day populations: Mbuti (Africans), Palestinians, Iranians (Middle Easterners), Armenians from the southern Caucasus, Papuans [24], Nganasans, Kets, Koryaks (Siberians), Karitiana (Native Americans), Irish, and Sardinians (Europeans).

We first explored cladality of population pairs using *qpWave* (Fig 5 and S2 Table). Specifically, we tested if one stream of ancestry from an ESEA surrogate is sufficient to model a Southeast Asian target population. We used a cut-off p-value of 0.05. We further tested 2-way and 3-way admixture models using *qpAdm*. We applied two criteria for defining plausible admixture models: a) the model is not rejected according to the chosen p-value cutoff; b) inferred admixture proportions ± 2 standard errors lie between 0 and 1 for all ancestry components. If a model meets these criteria, we consider the model as "fitting" or "passing" (S2

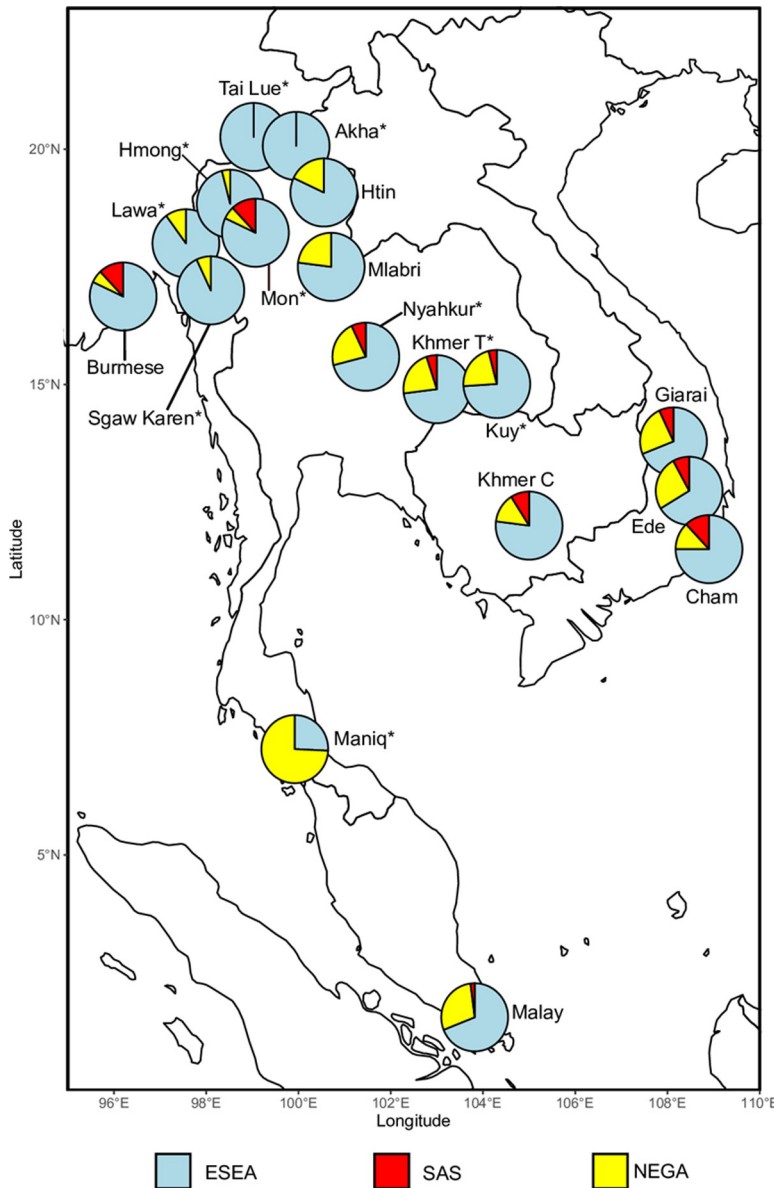

**Fig 5. An overview of admixture proportions estimated using *qpAdm*.** Admixture proportions were inferred using *qpAdm* with three groups of surrogates representing three ancestries: deeply diverged East Eurasian represented by Onge (NEGA), South Asian (SAS), and East and Southeast Asian (ESEA). Admixture proportions shown here were averaged across all models which passed our plausibility criteria. The map was plotted using an R package "rnaturalearth" (https://github.com/ropensci/rnaturalearth) with Natural Earth map data (https://www. naturalearthdata.com/). Asterisks after population names indicate that these populations are newly genotyped in this study.

Table), although we caution that the only secure interpretation of *qpWave* or *qpAdm* tests is in terms of model rejection, and not model fit [31]. For testing 2-way and 3-way admixture, we constructed models "ESEA + NEGA" and "ESEA + NEGA + SAS", respectively (Fig 5 and S2 Table).

Next, we tested more explicit demographic models using *qpGraph* [24]. *qpGraph* is an *f*-statistics-based tool which is used for testing demographic models in the form of a phylogenetic tree with pulse-like admixture events. *qpGraph* also reports optimized branch lengths and

admixture proportions. The aim of our *qpGraph* analysis was to investigate ancestry sources in a way that does not rely exclusively on available surrogate groups (like *qpAdm* does); the *qpAdm* framework cannot account for ancestry from "ghost" sources. We first constructed two skeleton graphs using different SAS surrogates, Coorghi (S2A Fig) and Palliyar (S2B Fig). Skeleton graph construction is detailed in Materials and Methods. The worst-fitting $f_4$-statistics predicted under the two skeleton graph models differed from the observed values by 2.43 and 2.24 SE intervals, respectively. These values are termed worst residuals or WR below. We then exhaustively mapped target ESEA populations on all possible edges (except for edge0 in S3 Fig) on the skeleton graphs. We modeled the target populations as unadmixed (33 models per target population per skeleton graph), 2-way admixed (528 models), and 3-way admixed (5,456 models). We compared models with different numbers of admixture sources using a log-likelihood difference cut-off of 10 log-units or a worst residual difference cut-off of 0.5 SE intervals (see exploration of appropriate cut-offs on simulated genetic data in [31]). For models with the same number of admixture sources, we used a log-likelihood difference cut-off of 3 log-units [32]. We also avoided models with trifurcations, i.e., when drift length on any "backbone" edge equals zero. A summary of *qpWave*, *qpAdm*, *qpGraph*, and SOURCEFIND results is presented in Table 1. Full results are shown in S2 Table (*qpWave* and *qpAdm*) and S3 Table (*qpGraph*). S3 Table shows all *qpGraph* models satisfying the log-likelihood difference criteria. Edge labels for the Coorghi and Palliyar skeleton graphs are shown in S3 Fig.

Akha, Sgaw Karen, and Lawa harbor ancestry from a Tibetan-related source (see a simplified rendering of the best-fitting admixture graph model for Lawa in Fig 6A, best-fitting graph models for all target groups in Figs 6 and S4, and full results in S3 Table). Akha was modeled as one stream of ancestry when Lahu was used as an ESEA surrogate in *qpWave* (S2 Table). Sgaw Karen requires an additional ancestry source from the Onge surrogate in *qpAdm* analysis (Fig 5 and S2 Table). The result agrees with the *qpGraph* analysis where Sgaw Karen was modeled as a mixture of a Tibetan-related and a Mlabri-related source (S3 Table and S4B Fig). Mlabri harbor a substantial proportion of deeply diverged East Eurasian ancestry (Figs 5 and

**Table 1. A summary of *qpAdm*, *qpGraph*, and SOURCEFIND admixture modelling results for the groups of interest.** Labels of groups genotyped in this study are italicized and marked with asterisks.

| population | n | language family | country | qpAdm, best model | qpGraph, best model | SOURCEFIND |
|---|---|---|---|---|---|---|
| Cambodian Khmer | 9 | Austroasiatic | Cambodia | ESEA + NEGA + SAS | Atayal + Mlabri + SAS | ESEA + SAS |
| Htin | 10 | Austroasiatic | Thailand | ESEA + NEGA | Mlabri | ESEA |
| *Khmer from Thailand** | 10 | Austroasiatic | Thailand | ESEA + NEGA or ESEA + NEGA + SAS | Mlabri + SAS | ESEA + SAS |
| *Kuy** | 10 | Austroasiatic | Thailand | ESEA + NEGA or ESEA + NEGA + SAS | Mlabri + SAS | ESEA + SAS |
| *Lawa** | 10 | Austroasiatic | Thailand | ESEA + NEGA | Tibetan + Mlabri | ESEA |
| *Maniq** | 9 | Austroasiatic | Thailand | ESEA + NEGA | Atayal + NEGA (after Onge) | - |
| Mlabri | 10 | Austroasiatic | Thailand | ESEA + NEGA | included in the skeleton graphs | - |
| *Mon** | 10 | Austroasiatic | Thailand | ESEA + NEGA + SAS | before Tibetan + Mlabri (ESEA source) + SAS | ESEA + SAS |
| *Nyahkur** | 9 | Austroasiatic | Thailand | ESEA + NEGA + SAS | Mlabri + SAS | ESEA + SAS |
| Cham | 10 | Austronesian | Vietnam | ESEA + NEGA + SAS | Atayal + Mlabri + SAS (western source) | ESEA + SAS |
| Ede | 9 | Austronesian | Vietnam | ESEA + NEGA + SAS | Mlabri + SAS | ESEA + SAS |
| Giarai | 11 | Austronesian | Vietnam | ESEA + NEGA + SAS | Mlabri + SAS | ESEA + SAS |
| Malay | 5 | Austronesian | Singapore | ESEA + NEGA or ESEA + NEGA + SAS | Atayal + Mlabri + SAS | ESEA + SAS |
| *Hmong** | 10 | Hmong-Mien | Thailand | ESEA + NEGA | before Atayal + Tibetan | ESEA |
| *Tai Lue** | 9 | Kra-Dai | Thailand | ESEA | before Dai/Mlabri + Mlabri | ESEA |
| *Akha** | 31 | Sino-Tibetan | Thailand | ESEA | Tibetan + Mlabri (ESEA source) | ESEA |
| Burmese | 6 | Sino-Tibetan | Myanmar | ESEA + NEGA+ SAS | Tibetan + Mlabri + SAS | ESEA + SAS |
| *Sgaw Karen** | 10 | Sino-Tibetan | Thailand | ESEA + NEGA | Tibetan + Mlabri | ESEA |

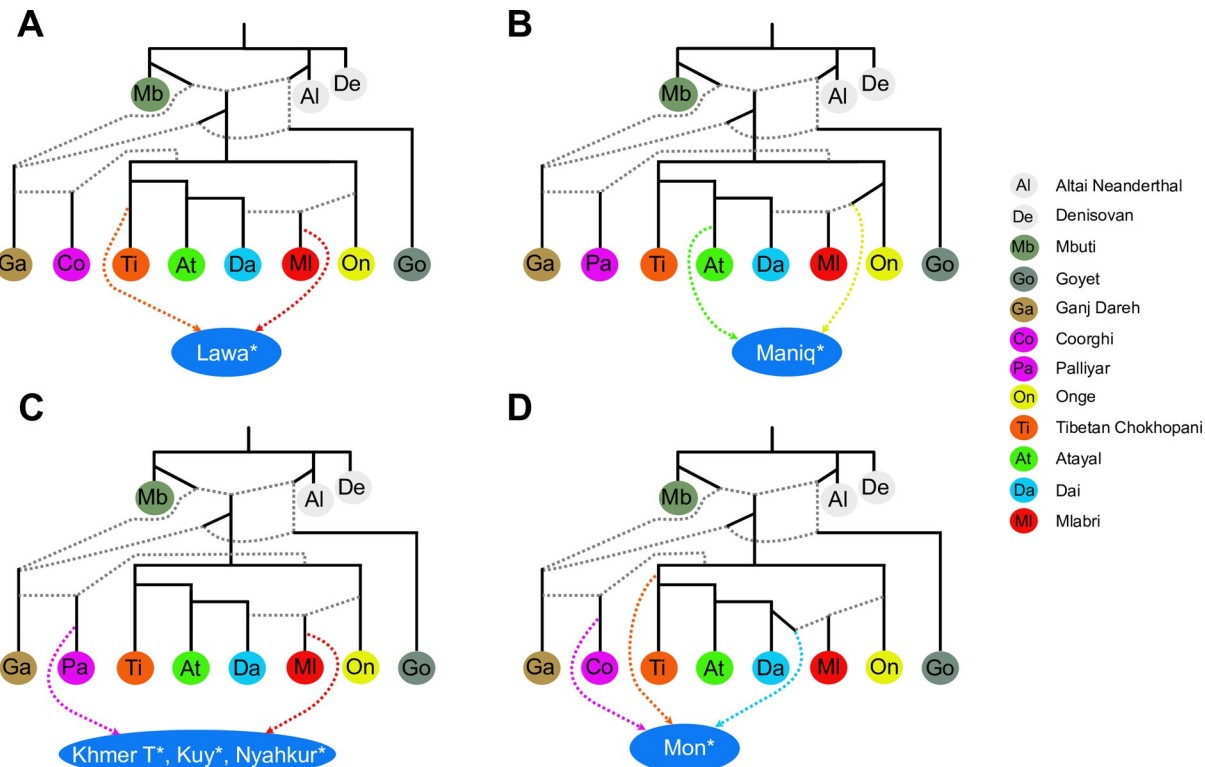

**Fig 6. An illustration of best-fitting *qpGraph* models.** Best-fitting *qpGraph* models for selected Southeast Asian target populations are presented: (**A**) Lawa, (**B**) Maniq, (**C**) Khmer from Thailand, Kuy, Nyahkur, and (**D**) Mon. Dashed lines represent admixture edges. Each target population was tested separately. *qpGraph* inferred the same sets of ancestry sources for Khmer from Thailand, Kuy, and Nyahkur. Therefore, we depict these three populations together in (**C**). Asterisks after population names indicate that these populations are newly genotyped in this study. Best-fitting models for the other target populations are shown in S4 Fig.

S2). An additional gene flow from deep sources (edge7 or edge8) to Karen on the Coorghi skeleton graph decreased the worst residual by ~0.5 SE intervals, but the inferred admixture proportion was close to zero (S3 Table); therefore, these additional edges could be an artifact. Lawa was modeled as Mlabri-related + Tibetan-related ancestry (Fig 6A and S3 Table).

Using *qpWave*, we were not able to model Hmong as cladal with any of our three standard ESEA surrogates (Atayal, Dai, and Lahu) (S2 Table). Then we tried to use Miao, a Hmong-Mien-speaking population from China, as an ESEA surrogate. We successfully modeled Hmong as Miao + Onge (S2 Table). The Hmong groups from Thailand and from Vietnam [16] are cladal according to *qpWave* (S2 Table). Our *qpGraph* analysis showed a low level of Tibetan-related ancestry (~2%) in Hmong (S4C Fig and S3 Table).

Htin was modeled as a sister group of Mlabri by *qpGraph* (S4D Fig and S3 Table). Both groups were modelled by *qpAdm* as having ESEA and Onge-related ancestry (Fig 5 and S2 Table). Maniq, a present-day hunter-gatherer Negrito group from Southern Thailand, has a predominant ancestry component derived from a deeply diverged East Eurasian group, with ~74% admixture proportion inferred by *qpAdm* (Fig 5 and S2 Table). The ESEA source for Maniq is Atayal-related, according to our *qpGraph* analysis (Fig 6B and S3 Table).

We detected South Asian admixture in MSEA populations which speak languages of different families: Austroasiatic (Khmer from Cambodia and Thailand, Kuy, Mon, and Nyahkur), Austronesian (Cham, Ede, and Giarai), and the Tibeto-Burman branch of the Sino-Tibetan family (Burmese). Austroasiatic-speaking groups in our study, Khmer from Cambodia and

Thailand, Kuy, Mon, and Nyahkur, harbor South Asian ancestry (9.4 ± 2.2%, 4.6 ± 1.3%, 4.3 ± 1.2%, 11.6 ± 1.3%, and 7 ± 1.6%, respectively), as inferred by *qpAdm* (Fig 5 and S2 Table). Khmer from Thailand, Kuy, and Nyahkur demonstrated similar genetic makeups (Fig 6C and S2 and S3 Tables). Using *qpGraph*, we also found Atayal-related ancestry in Cambodian Khmer (S4E Fig and S3 Table) and Tibetan-related ancestry in Mon, and these ancestry sources are rare in other Austroasiatic speaking populations (Fig 6D and S3 Table). There are four Austronesian-speaking populations included in this study: Cham, Ede (Rade), and Giarai (Jarai) from Vietnam [16], and Malay from Singapore [33]. *qpAdm* and *qpGraph* results revealed South Asian ancestry in all four Austronesian-speaking groups: 11.6 ± 2.5%, 7.5% ± 2.1, 7.4 ± 2.0%, and 2.1% in Cham, Ede, Giarai, and Malay, respectively, as inferred by *qpAdm* (S4F–S4H Fig and S2 and S3 Tables). Atayal is an Austronesian-speaking group from Taiwan, the homeland of Austronesian languages [34]. Using *qpGraph*, we did not detect Atayal-related ancestry in Ede and Giarai (S4G Fig and S3 Table), while that ancestry was found in Cham and Malay (S4F and S4H Fig). We also found Mlabri-related ancestry in all four Austronesian-speaking populations (S4F–S4H Fig and S3 Table). Both *qpAdm* and *qpGraph* analyses indicated South Asian ancestry in Burmese: e.g., ~12 ± 1.6% inferred by *qpAdm* (Figs 5 and S4I and S2 and S3 Tables). Burmese harbor ancestry from Tibetan-related + Mlabri-related + South Asian sources according to a best-fitting graph model (S4I Fig and S3 Table).

In order to test the genetic connection between Austronesian and Kra-Dai-speaking populations, we explored ancestry sources for Kra-Dai-speaking populations from China (Dong, Dong Hunan, Gelao, Li, Maonan, Mulam, and Zhuang from Wang et al., 2021 [35]), Vietnam (Boy, Colao, Lachi, Nung, Tay, and Thai from Liu et al., 2020 [16]), and Thailand (Tai Lue from this study) using *qpGraph*. We found that most Kra-Dai-speaking populations from China and Vietnam harbor Tibetan-related and Atayal-related ancestry (S4J Fig and S3 Table).

## Inferring sources of South Asian ancestry and admixture dates using haplotype-based analyses

For verifying our results by an independent method and for adding a "temporal dimension" to our work, we inferred sources of SAS ancestry in MSEA populations and dates of admixture events using haplotype-based methods, SOURCEFIND [36] and fastGLOBETROTTER [37,38], respectively. Both methods require inputs from ChromoPainter [39], a software which "paints" chromosomes of recipient individuals with haplotypes from donor populations. We first explored haplotype sharing within populations by allowing a recipient individual to receive haplotypes from all other populations and other individuals from the same population. Maniq and Mlabri display high levels of haplotype sharing within populations (90% and 88% of their genomes, respectively, was covered by haplotypes from donors in the same population) (S5 Fig), and that is a signature of a recent intense genetic drift. Considerable genetic drift in Maniq and Mlabri is also suggested by the ADMIXTURE results since both populations are modelled as having unique ancestry components at a ~100% level (Fig 3). For this reason, we excluded Maniq and Mlabri from further haplotype-based analyses. For SOURCEFIND and fastGLOBETROTTER analyses, we removed target populations from the list of donors, i.e., we did not allow target populations to share haplotypes neither with other target populations, nor within the same population.

SOURCEFIND infers population structure of target populations as a mixture of surrogate donor populations. According to the SOURCEFIND results, a predominant ancestry component for most MSEA populations in our study was represented by Kinh as a surrogate, except for Hmong, which was modeled as mixture of two other Hmong-Mien speaking populations, Miao and She (Fig 7A). According to this analysis, Akha received gene flows from various

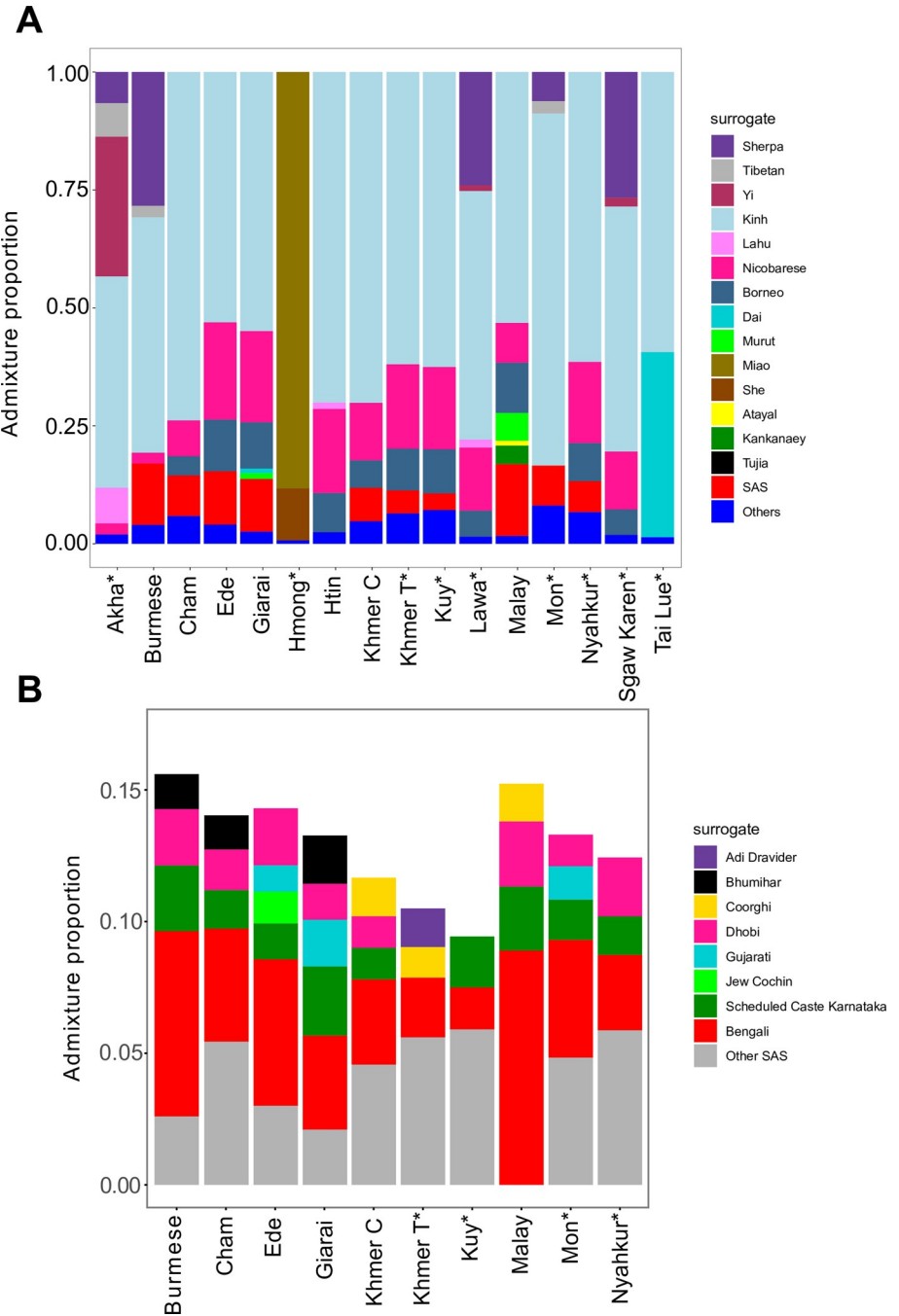

**Fig 7. Admixture proportions inferred using SOURCEFIND.** (**A**) Admixture proportions from all surrogates contributing at least 1% of target population's ancestry. The sum of ancestry proportions from all other surrogates is labeled as Others. (**B**) The same results are shown for SAS surrogates only. Asterisks after population names indicate that these populations are newly genotyped in this study.

Sino-Tibetan surrogates, namely Yi, Tibetan, Lahu, and Sherpa. Sherpa contributes more than 20% of ancestry in Burmese, Sgaw Karen, and Lawa. About 10% of Mon ancestry is derived from Sherpa and Tibetans (Fig 7A). SOURCEFIND also detected SAS admixture in Burmese, Khmer from Cambodia and Thailand, Cham, Ede, Giarai, Kuy, Malay, Nyahkur, and Mon

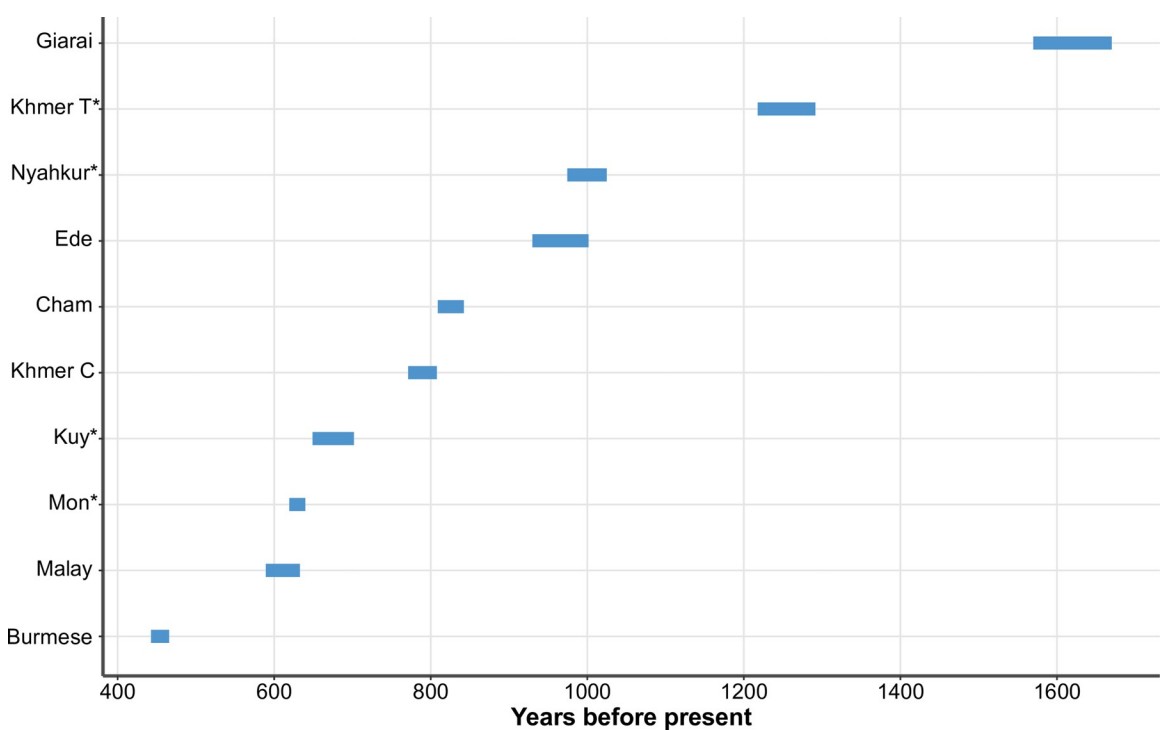

**Fig 8. Dates of South Asian admixture events inferred by fastGLOBETROTTER.** The plot shows 95% confidence intervals of admixture dates estimated from 100 bootstrap replicates. Asterisks after population names indicate that these populations are newly genotyped in this study.

(Fig 7A and 7B). The SAS signal is undetectable in other target populations (Fig 7A). These results are consistent with other analyses in this study. SOURCEFIND found eight SAS donors, which contribute more than 1% of ancestry to target populations (Fig 7B). Locations of SAS surrogates on the map are illustrated in S6 Fig. Among the eight SAS sources, Bengali emerged as the most prominent ancestry source for all the target populations. We caution that Bengali is the only SAS surrogate which harbors detectable East Asian ancestry (S7 Fig). This fact may lead to excessive haplotype sharing between Bengali and the MSEA target populations as compared to the other SAS surrogates.

We further dated SAS admixture events in the target populations using fastGLOBETROT-TER. The software inferred Bengali as the South Asian source in Malay, Cham, Ede, Giarai, Khmer from Cambodia and Thailand, Kuy, and Nyahkur, while the inferred South Asian surrogate in Burmese and Mon was Dhobi. The admixture dates (95% confidence intervals) in Giarai and Khmer from Thailand are older than 1000 years before present (YBP). The dates for Nyahkur and Ede are around 1000 YBP. For most populations, the SAS admixture was dated between 500–1000 YBP. Burmese demonstrated the youngest dates, 443–466 YBP (95% confidence interval) (Fig 8).

## Discussion

Indian culture was long established in MSEA and influenced formation of early states in the region during the first millennium CE [6]. Previous studies reported South Asian admixture in few populations from Southeast Asia [11–13]. Some studies analyzed the same or similar populations as those in the current study, but did not focus on South Asian admixture and did not report it [16,40,41]. In this study, we systematically explored South Asian admixture in

present-day Southeast Asian populations. We also investigated other aspects of genetic history in the region. Our results were consistent across various methods used (PCA, ADMIXTURE, $f_3$-statistics, qpAdm, qpGraph, SOURCEFIND, and fastGLOBETROTTER). For all target MSEA groups, we found just one or few admixture graph models that fitted the data significantly better than ca. 6000 other models we tested for each target group. qpAdm and qpGraph results agreed: adding a South Asian-related admixture edge never improved qpGraph model fits significantly when a 3-way qpAdm model with South Asian admixture was implausible according to our criteria. The presence of South Asian ancestry in Southeast Asian populations is also confirmed by SOURCEFIND, a haplotype-based analysis. Apart from South Asian admixture, we also explored other ancestry sources in various Southeast Asian populations. Below we discuss the genetic makeup of each population analyzed in this study.

Akha shares haplotypes with various Tibeto-Burman-speaking populations. Among the Tibeto-Burman surrogates (Lahu, Sherpa, Tibetan, and Yi), Yi contributes haplotypes to Akha more than other surrogates. In contrast, other target populations with Tibeto-Burman ancestry in this study receive haplotypes from Sherpa more than from other Tibeto-Burman surrogates (Fig 7A). Both qpGraph and SOURCEFIND detected a Tibeto-Burman-related signal in an Austroasiatic-speaking population Lawa (Figs 6A and 7A and S3 Table). Lawa likely got Tibetan-related ancestry via Sgaw Karen. Around 1850, Sgaw Karen started migrating from present-day Myanmar to the region that was once exclusively occupied by Lawa [42]. There are villages where both Lawa and Sgaw Karen live alongside each other [43], and intermarriage between the two groups became more common recently [44]. The genetic interaction between Karen and Lawa was also observed by Kutanan et al. [13]. SOURCEFIND modeled Hmong as a mixture of Miao and She as surrogates (Fig 7), two Hmong-Mien speaking populations from China. Our qpWave result demonstrates that Hmong from Thailand and Hmong from Vietnam are cladal (S2 Table), which suggests genetic continuity of Hmong populations from the two countries.

Htin can be modeled by qpGraph as a sister group of Mlabri (S4D Fig and S3 Table). Both Mlabri and Htin languages belong to the Khmuic branch of the Austroasiatic family [1]. A previous study showed that Mlabri has a genetic profile similar to early Neolithic individuals from mainland Southeast Asia [5]. The best-fitting qpGraph models for Maniq, a mainland Negrito group, incorporate 2-way admixture between an Atayal-related source and an Onge-related source, with a predominant genetic contribution from the latter source (Fig 6B and S3 Table). Even though Maniq speak an Austroasiatic language, a surrogate for their ESEA source picked up by qpGraph was Atayal, an Austronesian-speaking population (Fig 6B and S3 Table). Maniq may harbor Atayal-related ancestry from Austronesian-speaking populations in Southern Thailand (where they reside) or from Malaysia nearby. The interaction between Maniq and Malay is reflected by numerous Malay loanwords in the Maniq language [45]. Even though we cannot find a better admixture graph model for Maniq, we do not conclude that the Atayal-related source is the only ESEA source in Maniq as Atayal is not the only plausible ESEA source in our qpAdm analysis of Maniq (S2 Table).

Akha, Lawa, Karen, Hmong, and Htin were officially recognized as hill tribes by the Hill Tribe Development and Welfare Programme of the Department of Public Welfare in Thailand [21]. Maniq and Mlabri are the last two hunter-gatherer groups in Thailand [45]. We failed to detect South Asian ancestry in these seven populations. The result is consistent with a recent study by Kutanan et al. [13], which investigated eight hill tribes from Thailand and detected no South Asian admixture. These populations reside in remote areas and have received minimal influence from Indian culture, thus their ancestors likely had minimal contact with South Asian populations who migrated to the region in the past.

A Malay group from Singapore was modeled by *qpGraph* as a 3-way admixture involving sister groups of Atayal, Mlabri, and South Asian populations (S4H Fig and S3 Table). Malay is an Austronesian language. It is not surprising that the Malay harbor some ancestry from a source related to Atayal, an Austronesian-speaking population from Taiwan. A previous study found admixture from an Austroasiatic-speaking population in Austronesian populations from Indonesia [5]. We also detected the same signal in Malay, which is represented by ancestry from a sister group of Mlabri (S4H Fig and S3 Table). SOURCEFIND also reveals the presence of South Asian admixture in Malay (Fig 7). An earlier work by Kutanan et al. [13] also reported a similar genetic makeup of a Malay-speaking group from Thailand. The date of South Asian admixture in Malay inferred by fastGLOBETROTTER is around 600 YBP (Fig 8).

Cham, Ede, and Giarai are Austronesian-speaking populations from Vietnam. Using *qpGraph*, we were able to confirm the Atayal-related ancestry in Cham, but that signal was not detected in the cases of Ede and Giarai (S4F and S4G Fig and S3 Table). The results are consistent with a previous study by Liu et al. 2020 [16], which supports the spread of Austronesian languages by cultural diffusion in Ede and Giarai. West Eurasian-associated Y-haplogroups (R1a-M420 and R2-M479) were observed at low frequencies in Ede (8.3% and 4.2%) and Giarai (3.7% and 3.7%) by Machold et al. 2019 [10], and low frequencies of Y-haplogroups R-M17 (13.6%) and R-M124 (3.4%) were found in Cham by He et al. 2012 [46]. The *qpAdm*, *qpGraph*, and SOURCEFIND methods consistently infer South Asian ancestry in Cham, Ede, and Giarai (Figs 5, 7, S4F, and S4G and S2 and S3 Tables), but admixture dates in these populations are widely different (Fig 8).

In this study, we generated new data for Austroasiatic-speaking Khmer from Thailand. Khmer is the official language of Cambodia, and Cambodian Khmer (Cambodians) is the majority ethnic group in Cambodia [1]. Our admixture graph modeling showed that Khmer from Thailand and Cambodia harbor two ancestry sources in common: a Mlabri-related source and South Asian ancestry (Figs 6C and S4E and S3 Table). Low frequencies of West Eurasian-associated Y-haplogroups R1a1a1b2a2a (R-Z2123) and R1a1 were reported in Khmer from Thailand (3.4%) [9] and Cambodia (7.2%) [7], respectively. The best-fitting admixture graph model for Khmer from Cambodia includes additional ancestry from an Atayal-related (i.e., Austronesian) source (S4E Fig and S3 Table). Khmer from Cambodia plausibly received this ancestry via Cham due to a long-lasting interaction between the ancient Cambodian and Champa Kingdoms [6]. Cham is also the largest ethnic minority in Cambodia today [1]. Haplotype-based analysis SOURCEFIND also confirms South Asian admixture in Khmer from Thailand and Cambodia (Fig 7). The date of the South Asian admixture event is older in Khmer from Thailand (1218–1291 YBP) than in Khmer from Cambodia (771–808 YBP) (Fig 8), but both dates lie within the Angkorian period (9th - 15th century CE) [2]. All our analyses indicate South Asian admixture in Kuy (Figs 5, 6C and 7 and S2 and S3 Tables). Kutanan et al. 2019 [9] reported the presence of a West Eurasian-associated Y-haplogroup R1a1a1b2a1b (R-Y6) in Kuy at a low frequency (5%). fastGLOBETROTTER estimates the date of South Asian admixture in Kuy between 649 and 702 YBP (Fig 8). Even though the history of Kuy is not well known, a linguistic study [23] suggested a long-lasting contact between Kuy and Khmer starting before the Angkorian period.

We consistently inferred Tibetan-related ancestry in Burmese using q*pGraph* and SOURCEFIND analyses (Figs 7 and S4I and S3 Table). We found that ancestry component in all Tibeto-Burman-speaking populations in our study (Figs 7, S4A, S4B and S4I and S3 Table). The South Asian admixture in Burmese was dated between 443 and 466 YBP, which falls into the period of the First Toungoo Empire in Myanmar history [22].

Mon and Nyahkur languages belong to the Monic branch of the Austroasiatic family [1]. Our *qpAdm*, *qpGraph*, and SOURCEFIND analyses found South Asian ancestry in both

populations (Figs 5, 6C and 6D and S2 and S3 Tables). A previous Y-chromosome study [9] reported low frequencies of various West Eurasian-associated haplogroups, such as J (5%) and R (16%), in Mon, and of haplogroup J2a1 (J-L26) (5%) in Nyahkur. The higher frequencies of West Eurasian-associated Y-haplogroups in Mon correspond to the higher South Asian admixture proportion found in Mon as compared to Nyahkur (12% in Mon vs. 7% in Nyahkur, as inferred using *qpAdm*). The Nyahkur group is possibly a remnant of an ancient Monic-speaking population from the Dvaravati period located within present-day Thailand [47]. The South Asian admixture dates in Nyahkur also fall within the Dvaravati period that lasted from the 6th to the 11th centuries CE [48] (Fig 8). The inferred South Asian admixture date in Mon is around 600 YPB (Fig 8), which fits the Mon Independent Ramanya Polity period (1300–1539) in present-day lower Myanmar [22] and is close to a previous estimate by Kutanan et al. [13]. Mon harbors additional ancestry from a Tibetan-related source (Figs 6D and 7 and S3 Table), but this ancestry is missing in Nyahkur (Figs 6C and 7 and S3 Table). Mon probably received Tibetan-related ancestry via interactions with Sino-Tibetan-speaking populations in Myanmar. After the Burmese army from the first Toungoo kingdom conquered the Mon state in present-day lower Myanmar, a Burmese king Tabinshweihti established a Mon city Pegu (Bago) as the capital city of the first Toungoo kingdom in 1539 [22]. There is some debate about the origin of Mon in the Lamphun province (where they were sampled for our study): whether they are direct descendants of people from the ancient Mon states in present-day Thailand (Dvaravati or Haripunjaya; the present-day Lamphun was a part of Haripunjaya until 1292 CE [6]), or their ancestors migrated from Myanmar in the last few hundred years [45]. Our results favor the latter possibility due to the Tibetan-related genetic component found in Mon from Lamphun, which may reflect interaction with Burmese or other Tibeto-Burman-speaking populations in Myanmar where the density of such populations is much greater than in Thailand [1]. The South Asian admixture date in Mon also fits the period of the ancient Mon state in present-day Myanmar. Furthermore, the Tibetan-related ancestry is absent in Nyahkur, another Monic-speaking population from Thailand.

Atayal-related ancestry was found in most Kra-Dai-speaking populations in China and Vietnam, according to our analysis (S3 Table). Besides the Kra-Dai speakers, we were able to detect Atayal-related ancestry only in Austronesian-speaking populations (Malay, Cham) or in non-Austronesian-speaking populations which have historical evidence of interactions with Austronesian-speakers such as Maniq and Khmer from Cambodia (S3 Table). The genetic link between Austronesian-speaking and Kra-Dai-speaking populations in our modeling may reflect a deep relationship of the two language families as suggested by the Austro-Tai hypothesis [18]. Tai Lue is one of the Dai ethnic groups originating in South China [49]. The ancestors of the Tai Lue volunteers in our study migrated to Thailand less than a century ago from Myanmar. Cladality of Tai Lue with all three ESEA surrogates was not rejected using *qpWave* (S2 Table). However, *qpGraph* modeling supported a more complex model for Tai Lue: 2-way admixture between a source close to Dai and either a Mlabri-related source or a source diverging before Atayal (S4J Fig and S3 Table). The result suggests that after the migration from China, Tai Lue admixed with local MSEA populations, or that the genetic makeup of the Dai group that gave rise to the Tai Lue group studied here was different from the Dai groups sampled previously [50]. SOURCEFIND analysis shows that Tai Lue receives much more haplotypes from the Dai surrogate as compared to other target populations in this study (Fig 7A).

We attempted to identify South Asian sources in SEA populations using a haplotype-based method SOURCEFIND. There are 8 out of over 50 South Asian surrogates contributing at least 1 percent to SEA populations (Fig 7). These populations come from different parts of India and Bangladesh (S6 Fig). George Cœdès suggested that all regions of India had some contribution to the spread of Indian culture to Southeast Asia, but the influence of Southern

India was predominant [6]. fastGLOBETROTTER inferred a wide range of dates for the South Asian admixture in Southeast Asian populations, most of them after the first millennium CE. Our results suggest that there may have been multiple waves of diverse South Asian populations that migrated to MSEA. Cœdès suggested that the Indian expansion involved various SEA regions and lasted several centuries [6]. Ancient samples should provide further insight into the timeframe of the South Asian migration to MSEA.

Our study revealed substantial South Asian admixture in various populations across Southeast Asia (~2–16% as inferred by *qpAdm*). For four ethnolinguistic groups genotyped in our study (Khmer from Thailand, Kuy, Mon, and Nyahkur) and four groups from published data (Cambodian Khmer, Cham, Ede, and Giarai [16,40]) South Asian admixture was not reported before in the literature, and we report it here. Thus, we for the first time demonstrate South Asian admixture in populations from Cambodia and Vietnam, extend earlier results detecting this ancestry component in Thailand [13], and confirm detection of this ancestry component in Myanmar and Singapore [12]. A recent study [13] also observed South Asian ancestry in various groups of Thai, a Malay group from Thailand, and Mon from different locations in Thailand other than in the current study. Prior to our study, Mon was the only known Austroasiatic-speaking population in Mainland Southeast Asia with confirmed South Asian admixture [13]. We also detected South Asian ancestry in Mon (from different location other than in Kutanan et al., 2021 [13]) and four other Austroasiatic-speaking populations in Thailand and Cambodia (three newly genotyped populations and one from published data) which inherited languages and culture from ancient Indian-influenced states. The South Asian admixture dates inferred in our study fit the historical context of ancient Indian-influenced states. In concordance with an earlier study [13], we failed to detect South Asian admixture in relatively isolated "hill tribes" and in present-day hunter-gatherer groups from Thailand. Consequently, the spread of Indian influence in the region was accompanied by movement of people from India and was not a result of cultural diffusion only. The diversity of South Asian surrogates and inferred admixture dates indicates that there were multiple waves of South Asian populations migrating to MSEA.

In the current study, we analyzed populations from six branches of the Austroasiatic language family: 1) Khmuic (Htin and Mlabri), 2) Katuic (Kuy), 3) Palaungic (Lawa), 4) Monic (Mon and Nyahkur), 5) Khmeric (Khmer from Thailand and Cambodia), and 6) Aslian (Maniq) [1]. Kutanan et al. 2021 [13] investigated populations from four branches of the Austroasiatic language family in Thailand and revealed three genetic clusters, which fitted rather well with linguistic branches: 1) Khmuic and Katuic, 2) Paluangic, and 3) Monic. However, genetic profiles of some Austroasiatic-speaking populations in our analysis do not fit well with linguistic branches as they resemble populations from different branches, and there is genetic heterogeneity within linguistic branches. We also observed genetic differences between the Khmuic, Paluangic, and Monic groups in our study, i.e., we detected additional Tibetan-related ancestry in a Paluangic population (Lawa) as compared to Khmuic groups (Htin and Mlabri), while South Asian admixture in Monic populations (Mon and Nyahkur) distinguished them from the other two groups. Even though Khmuic and Katuic-speaking populations in Kutanan et al. 2021 [13] demonstrated a similar genetic makeup, genetic profiles the of Khmuic and Katuic-speaking groups in our study are distinct as a Katuic group (Kuy) harbors South Asian admixture, while that ancestry is missing in Katuic groups from Kutanan et al., 2021 [13] (Bru and Soa) and in Khmuic groups from both studies. Kuy's genetic profile is similar to that of Khmer from Thailand and Nyahkur, Austroasiatic-speaking populations from the Khmeric and Monic branches, respectively. We also observed genetic heterogeneity within linguistic branches of the Austroasiatic family, reflecting different interaction with other populations. Tibetan-related ancestry is present in Mon, but the ancestry is absent in

Nyahkur, another Monic-speaking group in our study. Khmer from Cambodia harbor Atayal-related admixture, but that ancestry is lacking in Khmer from Thailand. Maniq, an Aslian-speaking population, is distinct from other Austroasiatic-speaking populations, as their predominant genetic component is from an Onge-related deeply diverged East Eurasian lineage.

Some previous studies suggested a genetic link specifically between Austronesian-speaking and Kra-Dai- speaking populations, relying on $f_3$- or $f_4$-statistics [13,51,52]. We for the first time used *qpGraph* to directly estimate Atayal-related ancestry in dozens of Kra-Dai-speaking populations from China and Vietnam (~3–38% as inferred by *qpGraph*). Our results strengthen the previous results that Austronesian and Kra-Dai-speaking populations share a component of ancestry unique to those two groups.

Even though several recent studies explored genetic history of various populations from Mainland Southeast Asia [12,13,16,53], many populations in the regions are still uninvestigated, especially populations from Myanmar, Laos, and Cambodia. There were only few genome-wide studies on ancient people in the region [4,5]. Further data from present-day and ancient groups are expected to provide further insights into the genetic population history of Mainland Southeast Asia.

## Materials and methods

### Ethics statement

The study was approved by the Ethic Committee of Khon Kaen University.

### Sampling

Sample collection and DNA extraction for all new Thailand populations in this study apart from Akha was described in previous studies [9,14,41,54–56]. Saliva samples were obtained from volunteers who signed informed consent and who resided in four Akha villages in the Chiang Rai province, Thailand. We performed DNA extraction as described elsewhere [57]. See a list of individuals for whom genetic data is reported in this study in S4 Table.

### Dataset preparation

Diploid genome-wide SNP data was generated using the Affymetrix Human Origins SNP array [24]. We merged the new data with published ancient and present-day worldwide populations (S1 Table) using PLINK v.1.90b6.10 (https://www.cog-genomics.org/plink/). We first combined all present-day populations and applied a per site missing data threshold of 5% to create a dataset of 574,131 autosomal SNPs. We then added data from ancient populations. We used this dataset for all analyses except for ADMIXTURE.

### PCA

The principal component analysis (PCA) was performed using PLINK v.1.90b6.10 on selected populations (S1 Table) from the following regions: Central, East, Southeast, and South Asia, Andamanese Islands, Siberia, and Europe.

### ADMIXTURE

We performed LD filtering using PLINK v.1.90b6.10 with the following settings: window size = 50 SNPs, window step = 5 SNPs, $r^2$ threshold = 0.5 (the PLINK option "—indep-pairwise 50 5 0.5"). LD filtering produced a set of 270,700 unlinked SNPs. We carried out clustering analysis using ADMIXTURE v.1.3 (https://dalexander.github.io/admixture/download.html), testing from 8 to 14 hypothetical ancestral populations (K) with tenfold cross-validation and

ran 5 algorithm iterations for each value of K. We selected K = 12 for presentation as CV errors were not significantly different for K from 12 to 14 (S8 Fig), and we chose the simplest of those models. We further ran up to 30 algorithm iterations for K = 12 and ranked them by model likelihood. We chose an iteration with the highest model likelihood.

## Outgroup $f_3$-statistics

We computed $f_3$-statistics [24] using *qp3Pop* v.420, a software from the ADMIXTOOLS package (https://github.com/DReichLab/AdmixTools). We ran $f_3$(Mbuti; X, test group), where X stands for East Asian surrogates (Han or Dai) or South Asian surrogates (Brahmin Tiwari or Coorghi). The test groups are various ESEA populations.

## qpWave and qpAdm

We used *qpWave* v.410 and *qpAdm* v.810 from the ADMIXTOOLS package. We used the following populations as outgroups ("right populations") for all *qpWave* and *qpAdm* analyses: Mbuti (Africans), Palestinians, Iranians (diverse Middle Easterners), Armenians (Caucasians), Papuans [24], Nganasan, Kets, Koryaks (diverse Siberians), Karitiana (Native Americans), Irish, and Sardinians (diverse Europeans). We used Atayal, Dai, and Lahu as ESEA surrogates and Onge as a surrogate for the deeply diverged East Eurasian hunter-gatherers. We used 55 different populations as alternative South Asian surrogates (S2 Table).

We tested cladality of an MSEA population and an ESEA surrogate using *qpWave*. We used a cut-off p-value of 0.05. We also performed 2-way and 3-way admixture modeling using *qpAdm*. 2-way admixture was modeled as "target population = ESEA surrogate + NEGA (Onge) surrogate", and 3-way admixture was modeled as "target population = ESEA surrogate + NEGA (Onge) surrogate + SAS surrogate". We applied two criteria for defining plausible admixture models: a) the model is not rejected according to the chosen p-value cutoff; b) inferred admixture proportions ± 2 standard errors lie between 0 and 1 for all ancestry components.

## qpGraph

We used qpGraph v.6412 from the ADMIXTOOLS package with the following settings: outpop: NULL, blgsize: 0.05, lsqmode: NO, diag: 0.0001, hires: YES, initmix: 1000, precision: 0.0001, zthresh: 0, terse: NO, useallsnps: NO. We used the following criteria to select best-fitting models. Models with different numbers of admixture events were compared using a log-likelihood difference cut-off of 10 log-units or a worst residual difference cut-off of 0.5 SE intervals [31]. We used a log-likelihood difference cut-off of 3 log-units for comparing models with the same number of parameters [32].

We started building the skeleton admixture graph with the following five populations: Denisovan and Altai Neanderthal (archaic humans), Mbuti (African), Atayal (East Asian), and Goyet (ancient West European hunter-gatherer). A best-fitting model is illustrated in S9 Fig. We fixed the Neanderthal-related (node nA in S9 Fig) admixture proportion in non-Africans at 3%. Goyet requires extra admixture from this Neanderthal-related source. When this admixture edge was missing, the worst $f_4$-statistic residual increased from 2.13 to 4.56. We further mapped additional populations on the graph, one at a time. We mapped a new population on all possible edges on the graph as unadmixed, 2-way, and 3-way admixed. We mapped Onge on the 5-population graph (S9 Fig) and then Dai on the 6-population skeleton graph (S10 Fig). Best-fitting graphs including Onge and Dai are shown in S10 and S11 Figs, respectively.

We further mapped an ancient Iranian herder individual from Ganj Dareh (I1947 [25]). A best-fitting model for this individual is a 2-way mixture between a putative West Eurasian

source and a basal Eurasian source (S12 Fig). Basal Eurasian admixture in ancient groups from Iran was reported in a previous study [58]. Mlabri can be modeled as ESEA + Onge-related sources (S13 Fig), which is consistent with a previous study [5].

We mapped South Asian populations, Coorghi or Palliyar, on the graph in S13 Fig. Both populations can be modeled as a 2-way mixture between ancient Iranian-related and deep-branching East Eurasian sources (S14 Fig). The positions of the deep East Eurasian source for Coorghi and Palliyar are slightly different, but both are among the deepest East Eurasian branches.

We added an ancient Tibetan individual, Chokhopani from Nepal (S1 Table), as the last population on the skeleton graphs. The best-fitting model for this individual was an unad-mixed branch in the ESEA clade before the divergence of Atayal (S2 Fig). The total numbers of SNPs used for fitting the skeleton graphs with Coorghi and Palliyar were 311,259 and 317,327, and the worst $f_4$-statistic residuals were 2.43 and 2.24 SE, respectively.

We mapped present-day target populations on all possible edges (except for edge0 in S3 Fig) on the skeleton graphs as unadmixed, 2-way admixed, and 3-way admixed. In total, we tested 6,017 models per target population per skeleton graph.

## Haplotype-based analyses

We phased the world-wide dataset using SHAPEIT v.2 (r900) (https://mathgen.stats.ox.ac.uk/genetics_software/shapeit/shapeit.html) with 1000 Genomes Phase 3 genetic maps [59]. We then ran ChromoPainter v.2 [39] to generate inputs for SOURCEFIND and fastGLOBETROT-TER. We selected 18 target populations, 22 East Asian and Siberian surrogates, 2 Papuan sur-rogates, Onge, and 56 South Asian surrogates (S1 Table). We ran ChromoPainter v.2 with two different settings: 1) all populations were assigned as both donors and recipients, allowing hap-lotype sharing within populations; 2) all surrogates were assigned as donors and recipients, but target populations were assigned as recipients only. This means that target populations receive haplotypes only from surrogates, but not from their own population nor other target popula-tions. We first estimated switch rate and global mutation rate by running ChromoPainter v.2 on chromosomes 1 to 4 using 10 expectation-maximization iterations. We randomly selected 1/10 of individuals per population for this initial run (for populations with less than 10 individ-uals, we used 1 individual per population). We subsequently ran ChromoPainter v.2 using the switch rate and global mutation rate estimated at the previous step. All individuals of surrogate and target populations were used in this run. We used results of the first run (when all popula-tions were assigned as both donors and recipients) to estimate within-population haplotype sharing (S5 Fig). We used the output of the second run (when target populations were assigned as recipients only) for downstream analyses.

We inferred population structure of target groups using SOURCEFIND [37] with the fol-lowing settings: 1) allow up to 8 surrogates to contribute more than 0% ancestry for each itera-tion (num.surrogates: 8); 2) run 200,000 iterations in total (num.iterations: 200000); 3) discard the first 50,000 iterations as a burn-in (num.burnin: 50000); and 4) sample posterior admixture proportions every 5,000 iterations (num.thin: 5000). We averaged admixture proportions across 30 posterior samples and set a cutoff for admixture proportions at 1%. We ran fastGLO-BETROTTER [37,38] under default settings to date South Asian admixture events in MSEA groups. 95% confidence intervals for admixture dates were estimated relying on 100 bootstrap replicates.

## Supporting information

**S1 Fig. A biplot of** $f_3$(Mbuti; Coorghi, X) vs. $f_3$(Mbuti; Dai, X) (**A**), $f_3$(Mbuti; Coorghi, X) vs. $f_3$(Mbuti; Han, X) (**B**), and $f_3$(Mbuti; Brahmin Tiwari, X) vs. $f_3$(Mbuti; Dai, X) (**C**). Asterisks

after population names indicate that these populations are newly genotyped in this study.
(PDF)

**S2 Fig. Skeleton graphs used for the admixture graph mapping method.** We used the skeleton graphs to explore the genetic makeup of ESEA populations. We used different South Indian populations for two skeleton graphs: Coorghi in panel **A** and Palliyar in panel **B**.
(PDF)

**S3 Fig. Skeleton graphs used for admixture graph mapping, with edges numbered.** Coorghi was used as an Indian surrogate for skeleton graph (**A**) and Palliyar for skeleton graph (**B**).
(PDF)

**S4 Fig. Illustration of best-fitting *qpGraph* models.** Best-fitting *qpGraph* models for the following Southeast Asian target populations are presented: (**A**) Akha, (**B**) Sgaw Karen, (**C**) Hmong, (**D**) Htin, (**E**) Cambodian Khmer, (**F**) Cham, (**G**) Ede and Giarai, (**H**) Malay, (**I**) Burmese, (**J**) Tai Lue. Dashed lines represent admixture edges. Each target population was tested separately. *qpGraph* inferred the same sets of ancestry sources for Ede and Giarai. Therefore, we depict these two populations together in (**G**). Asterisks after population names indicate that these populations are newly genotyped in this study.
(PDF)

**S5 Fig. Haplotype sharing within populations.** The plot represents proportions of shared haplotypes within-population to total shared haplotypes. Haplotype-sharing was inferred using ChromoPainter v.2. Asterisks after population names indicate that these populations are newly genotyped in this study.
(PDF)

**S6 Fig. Geographical locations of South Asian surrogates.** Colored circles represent South Asian surrogates contributing at least 1% of ancestry in any MSEA target group. South Asian surrogates which contribute less than 1% of ancestry are labeled as Others. The map was plotted using an R package "rnaturalearth" (https://github.com/ropensci/rnaturalearth) with Natural Earth map data (https://www.naturalearthdata.com/).
(PDF)

**S7 Fig. East Asian admixture proportion in Bengali inferred using SOURCEFIND.**
(PDF)

**S8 Fig. Cross-validation (CV) error plot for the ADMIXTURE models.**
(PDF)

**S9 Fig. An initial skeleton admixture graph with 5 populations.**
(PDF)

**S10 Fig. The best-fitting model including Onge mapped on the 5-population skeleton graph (S9 Fig).**
(PDF)

**S11 Fig. The best-fitting model including Dai mapped on the 6-population skeleton graph (S10 Fig).**
(PDF)

**S12 Fig. The best-fitting model including an ancient Iranian herder form Ganj Dareh mapped on the 7-population skeleton graph (S11 Fig).**
(PDF)

**S13 Fig. The best-fitting model including Mlabri mapped on the 8-population skeleton graph (S12 Fig).**
(PDF)

**S14 Fig.** The best-fitting models including Coorghi (A) or Palliyar (B) mapped on the 9-population skeleton graph (S13 Fig).
(PDF)

**S1 Table. Information on reference populations used in this study and a list of populations used in haplotype-based analysis.**
(XLSX)

**S2 Table.** *qpWave* and *qpAdm* results.
(XLSX)

**S3 Table. All well-fitting *qpGraph* models.**
(XLSX)

**S4 Table. Metadata for newly genotyped present-day individuals.**
(XLSX)

## Acknowledgments

We thank all volunteers in Thailand who donated the samples for our study. We thank Phangard Neamrat, Jaeronchai Chuaychu, and Prateep Panyadee for assisting with sample collection.

## Author Contributions

**Conceptualization:** Piya Changmai.

**Data curation:** David Reich.

**Formal analysis:** Piya Changmai, Kitipong Jaisamut.

**Resources:** Piya Changmai, Jatupol Kampuansai, Wibhu Kutanan, Olga Flegontova, Angkhana Inta, Horolma Pamjav, David Reich, Pavel Flegontov.

**Supervision:** Pavel Flegontov.

**Visualization:** Kitipong Jaisamut, N. Ezgi Altınışık.

**Writing – original draft:** Piya Changmai, Pavel Flegontov.

**Writing – review & editing:** Piya Changmai, Kitipong Jaisamut, Jatupol Kampuansai, Wibhu Kutanan, N. Ezgi Altınışık, Olga Flegontova, Angkhana Inta, Eren Yüncü, Worrawit Boonthai, Horolma Pamjav, David Reich, Pavel Flegontov.

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
