## [Decision Letter · Decision Letter 0]

5 Mar 2021

Dear Dr Changmai,

Thank you very much for submitting your Research Article entitled 'Indian genetic heritage in Southeast Asian populations' to PLOS Genetics.

The manuscript was fully evaluated at the editorial level and by independent peer reviewers. The reviewers appreciated the attention to an important problem, but raised some substantial concerns about the current manuscript. Based on the reviews, we will not be able to accept this version of the manuscript, but we would be willing to review a much-revised version. We cannot, of course, promise publication at that time.

If you decide to revise the manuscript for further consideration at PLOS Genetics, please aim to resubmit within the next 60 days, unless it will take extra time to address the concerns of the reviewers, in which case we would appreciate an expected resubmission date by email to plosgenetics@plos.org.

[LINK]

We are sorry that we cannot be more positive about your manuscript at this stage. Please do not hesitate to contact us if you have any concerns or questions.

Yours sincerely,

Sandra Beleza

Guest Editor

PLOS Genetics

Gregory Barsh

Editor-in-Chief

PLOS Genetics

Editor: All reviewers think that this manuscript is on a topic of interest within the subject of Human origins and history of major human migrations and that has original results about the role of population movements in spreading cultural practices. However, they have made a number of significant points that should be addressed, as follows:

Reviewer's Responses to Questions

**Comments to the Authors:**

Reviewer #1: The authors address the potential genetic impact of Indian affiliated states in Mainland Southeast Asia (MSEA). These states emerged in the first millennium CE and had a massive cultural impact on MSEA still plain to see even today. The sample size is not mind boggling but ~120 well-chosen new samples from 10 populations seem to be quite sufficient for the task. The general result is clear. The authors identify Indian genetic ancestry in those MSEA populations who show also Indian cultural ancestry. Importantly, and interestingly, there seems no Indian genetic impact in populations who remained culturally isolated until recently. This is a clear and neat study showcasing an example of genes-culture co-migration.

Perhaps the limitation of the study is that it is a case study and thus difficult to generalize. But this is not to be held against the study as a) it is from many of such studies one would draw general patterns, and b) these results are interesting and relevant in their own right.

The paper would benefit from some reorganisation of the text and some amendments. Introduction already presents the results while it should identify the knowledge gap in the field and the aims of the study. The Results and also most of Discussion is a bit too dry. This latter point is for consideration only. If the authors agree they could try to write it up in a more smoother manner – like the very last paragraph of the Discussion. Figures could be made easier to follow and carry more information.

Minor comments

Line 37: “We found South Asian low-level admixture in various MSEA populations” better: “We found low-levels of South Asian admixture in various MSEA populations”

Line 82 – 91 This part of the Introduction section strikes as odd. It is indeed a summary of the results. In a way a slightly expanded abstract. Conventionally the introduction section should end with identifying the knowledge gap and the more specific aims of the study in closing (some of) this gap.

Figure 1 could be much better. Especially the zoom-in is uninformative in the current state. The background map should show more context. Both geographical, so that one could orientate, and populational – the geographic spread of the populations as opposed to only the sampling locations. The latter is not essential but nice to have.

Figure 2 – Could be better. For the main PCA it would be easier to read if the eight population group names would be spelled out on the figure. If possible indicate also the three main groups of Siberians. The zoom-in is very difficult to follow in the current form. It is easy to lose track in following the symbol shapes on the figure and in the legend. The line width of the symbols in the legend is too thin to actually see the colour. One possibility is using three-letter population codes instead of symbols on the plot itself. And define these in the supplementary table or in the legend. In this way it is easier to follow when one zooms into the plot. Also the colour code for the language groups in the zoom in panel should be shown on the figure.

ADMIXTURE analysis – the choice of K=12 based on highest model likelihood is interesting. In my experience LL values go up as one increases the value of K. There is of course variation so it may be that if one does only 5 replicates then by chance an LL of a run at K=12 is larger than that of runs at K=13. The choice of K is on one hand not perhaps critical as the results at any K are representation of population structure. But some K-s are more easily interpretable in terms of ancestry. What I’m getting at is for example the component maximized in Mlabri is probably there because of very small population size and hence high sharing within the group. In the current representation the figure might suggest a very different ancestry for this population. The same goes for Maniq. Without explanation perhaps a more intuitive K would be better. On the other hand I agree that bringing them out on the figure might make sense. But then there should be more explanation in the text.

This becomes even more important when one uses a % of a component to group populations. Did Mlabri and Maniq show SAS component at lower Ks? From the following analyses I draw that probably not. But still – using a percent of a component in ADMIXTURE analysis to group populations where the reason some populations may lack the component in question because they are maximized in their own component (due to high within-group sharing), can be misleading.

And why is there white space between Mlabri and Burmese on the figure?

Line 113: Probably “Figs 3 and 5”. And Fig 3 should show which populations are grouped into Thai 1-3. Does the individual-level variation in SAS component allow grouping populations based on SAS %?

Figure 4 (and figure 2 zoom in) The explanation of the colour scheme should be on the figure.

Line 122 – I’m a bit confused – on Fig 4 I seem to count more populations shifted toward SAS than are listed in the sentence.

From line 128 – This section feels more like Methods rather than Results. Since the qpXXX approaches form the main backbone of the paper – it probably is justified. And it reads well considering it is description of methods. So I’m not recommending doing anything specific, just pointing it out so the authors can decide if they want to change anything.

From line 175 – The paper starts to present results based on language affiliation of the populations under study. While this probably makes sense – it comes a little bit out of the blue. This decision should be explained. Better still justified by results obtained so far. For example this setup could be introduced already in Introduction which as it is now ends with summary of results instead.

Overall the results section, especially the qpXXX part could use a softer landing. As it is we go too deep too quickly. I think the readers of PLoS Genetics constitute a group of experts in modern population genetics approaches but a little more context would be helpful. Also for example one of the main results presented in the abstract – the lack of SAS related ancestry in isolated populations – is not easily found/explicitly presented in the main text, except in the last paragraph of Discussion.

Line 298: This wording “West Eurasian Y haplogroups R1a1a1b2a2a (R-Z2123) and R1a1 were reported in Khmer [29] and Cambodians [30], respectively.” Is a bit too vague. Are these the main Y hgs in these populations? Or is it just that they are present there as minor hgs. I think it would be better to make it more clear. The same construction “were reported in” is used in several places in the Discussion.

Line 355 – This section seems not part of the Kra-Dai section. More like conclusion? This paragraph is well written.

Reviewer #2: Changmai et al. in the manuscript “Indian genetic heritage in Southeast Asian populations” assembled a collection of population genetic samples across mainland Southeast Asia, and tested whether there is pervasive low level of South Asian ancestries in these populations. To this point, the authors did show a low (~2-16%) level of South Asian ancestry across majority of the populations they sampled. However, I feel that despite the stated motivation, the manuscript lacks a narrative around this thesis, and came off as a systematic application of qp-softwares of the populations they assembled. For example, there were descriptions and statements of the qp-analysis results, but no proposal of mechanisms. It seems the authors constantly shifted between a survey of the region with little depth, or a focus on the newly generated data from Thailand. As a result, the study lacked depth in application of population genetic analysis, and also fell short in illuminating knowledge regarding the formation of people around this region of the world.

Major comments:

1. The authors used really only one type of genomic information – the allele frequencies. Though branded as results that were “consistent across various methods used in this study”, they all more or less reflect the same underlying information. Then, there is really not much insight beyond what could be obtained from a naïve ADMIXTURE analysis of the sample. The authors would improve the impact of the paper by utilizing different sources of genomic information, such as LD-based analysis of admixture timing, or haplotype-based investigation of population structure. The genetic data are all based on modern individual; data quality is presumably high and should be amenable to these advanced analytic tools. The authors would also improve the narrative by commenting on how the set of qp-softwares differ from a vanilla ADMIXTURE / PCA analysis. What do they add above and beyond the vanilla analysis? How to decide which set of numbers to believe if ADMIXTURE results differ from qpAdm results?

2. In some cases, why not use the available aDNA data? There are places where I thought Hoabinhian sample could be helpful.

3. I had trouble understanding the significance of Thai1, 2, and 3. They appeared to be arbitrarily grouped based on average SAS ancestry from ADMIXTURE, but then were not really discussed afterwards. What is the significance of these three groups?

4. Throughout the paper, it was difficult to tell the contribution from the newly generated data. I think this will be clearer if the author focused on their newly generated data, and then clearly indicate when a finding based on newly generated data is supported/replicated from previously generated data from a nearby region.

5. In the Discussion, some groups had more insight than others (e.g. Austroasiatic and Kra-Dai). However, the discussion was no longer limited to proportion of South Asian ancestry, and seemed to distract from the main thesis as stated. The author may want to re-frame the central question on these specific language groups where they were able to (or were more interested to) derive more insight.

Other comments:

1. Description of Figure 2 in the Results: I think it is worth mentioning how the cline for CAS and for SIB are a little different, more consistent with the geographic expectation (i.e. SIB tending towards more European, CAS more in between Europeans and SAS).

2. Please consider changing the color scheme of Figure 2. I have trouble telling Munda from SAS, Onge from SIB, and NEGM from EUR.

3. Line 107 – how is K=12 chosen? In the Methods cross-validation was mentioned, but there was no explicit statement of how K=12 was chosen for presentation. If it is only be maximum likelihood, then increasing K should always increase likelihood. Are the estimated SAS ancestry consistent across Ks?

4. Line 111 and Figure 3 – it would be good to indicate which populations are the new data. Also label Thai1/2/3 on the figure as well (although see my comment above as I failed to see the purpose of creating these three subpopulations).

5. Figure 4 labels are actually different from the description in text from line 115-120.

6. Why not directly test admixture with f3 statistics? Figure 1 of the PCA appears to show quite a few choices one can pick from to model the two ancestral ancestry.

7. It would be good to state the intuition and general goals of qpWave, qpAdm, and qpGraph. Currently the authors only state methodically what was run (line 129-141, which could go into the Methods), without giving an intuition of what these methods are doing and why they are used here.

8. Line 145-148: it is difficult for me to parse the criteria for defining plausible admixture model. For example, “all simpler models should be rejected according to the chosen p-value cut-off” is confusing; shouldn’t the more parsimonious model be preferred? Perhaps the authors should clarify.

9. There appears to be no discussion of Figure 5 in the Results. They seem to be quite correlated to ADMIXTURE results (particularly looking at Thai 1/2/3), so what does this analysis actually add over ADMIXTURE?

10. Figure 6 is just the skeleton used for analysis, is it really worth a main figure?

11. Line 237-239: “adding a South Asian-related admixture edge never improved qpGraph model fits significantly when a 3-way model with South Asian admixture was rejected by qpAdm” – I do not understand the significance of this statement. Is this not expected?

12. Table S2 is extremely hard to parse – what is “conclusion”? What are columns N through V on the 3-way qpAdm tab? There was no description of the content of these supplementary tables.

13. Line 247 – “high worst residual” should be reworded.

14. Maybe a quarter of the Abstract seems to be material more suitable for the Introduction.

15. Line 37: “low-level South Asian admixture”.

Reviewer #3: The manuscript entitled “Indian genetic heritage in Southeast Asian populations” by Changmai et al. (PGENETICS-D-21-00062) reports the detailed analysis of the genetic structure and admixture history of diverse populations from Thailand, based on SNP array data for 119 individuals. The authors suggest that extensive admixture has occurred between South-East Asian and South Asian populations, possibly because of extensive migrations of South Asians during the “Indianization” of MSEA, as defined by Georges Coedès. The study design and analyses are relevant and straightforward. I appreciate that the authors discuss the accuracy of the methods used (qpGraph in particular), based on another study by the same lab. I nevertheless have a number of major comments.

- Novelty and robustness of the main conclusion. Evidence for a genetic heritage from South Asians in MSEA populations is not completely novel. It has been suggested by several previous studies, including a recent study based on SNP array data (Mörseburg et al., Eur J Hum Genet 2016 ; Kutanan et al., bioRxiv 2020). This is rightfully acknowledged by the authors; however, because this is the main conclusion of the current study, the authors should extend previous analyses, to strengthen their conclusion. Namely, if the evidence of South Asian gene flow to MSEA is strong, the time and precise geographical origins of this gene flow are not known. I suggest that the authors estimate the time of admixture between South Asian and MSEA populations, using e.g. ALDER. If estimated times are concordant with the historical period for Indian cultural influences in Southeast Asia, this could strengthen the authors’ conclusion. Moreover, I suggest that the authors determine which South Asian population is the best surrogate of South Asian ancestry in MSEA populations, using e.g. SOURCEFIND. There is considerable debate relating to the source of South Asian cultural influences in this region. This should be feasible given that South Asians are relatively diverse, and the authors already have access to 55 different populations from South Asia.

- The manuscript is lacking historical context. The authors should describe in more details in the introduction the current knowledge about the history of MSEA, and the hypotheses that remain to be tested, relating in particular to the “Indianization” of MSEA (see previous point). The authors could then highlight some of their results that may favor particular hypotheses (the larger Indian admixture proportions in coastal vs. northern Thailand, for example).

- Discussion about the results for populations speaking different linguistic families is very unbalanced. For Sino-Tibetan and Hmong-Mien speaking groups, there is virtually no discussion of the results, while those for Kra-Dai and Austroasiatic groups are extensively discussed. One option is to focus more the discussion on the main conclusion of the study. Can the authors speculate on the reasons why some groups admixed with South Asians and not others? Is geography the only driver of the patterns observed in Fig. 5? Languages do not seem to have been a barrier against gene flow, as evidence for admixture is found for Sino-Tibetan, Austronesian, Austroasiatic and Krai-Dai speakers.

- I do not think that the results support a “massive” migration of South Asians, as admixture proportions are ~5-10%. This is not comparable with the massive migrations inferred in Europe (Anatolian farmers, Pontic steppe herders), sub-Saharan Africa (Bantu speakers) or Oceania (Austronesian speakers).

Minor points

- The authors may clarify why some populations were chosen as surrogates for the skeleton graph. Why using (distant) Onge and not Maniq? Why using Coorghi and Palliyar as SAS surrogates?

- Please add linguistic groups in the legend of Fig. 1. Please remove the minus sign in “-PC1”.

- l. 111: clarify here why Thai samples are separated in three groups.

- L. 217: Please replace “form” by “from”

- L. 224: Please avoid ambiguous sentences such as “when we mapped the Thai1 population on the Coorghi skeleton”

- L. 234: Please replace “markup” by “makeup”

**Have all data underlying the figures and results presented in the manuscript been provided?**

Reviewer #1: Yes

Reviewer #2: Yes

Reviewer #3: Yes

PLOS authors have the option to publish the peer review history of their article (what does this mean?). If published, this will include your full peer review and any attached files.

Reviewer #1: No

Reviewer #2: No

Reviewer #3: No

---

## [Decision Letter · Decision Letter 1]

15 Sep 2021

Dear Dr Changmai,

Thank you very much for submitting your Research Article entitled 'Indian genetic heritage in Southeast Asian populations' to PLOS Genetics.

The revised manuscript was fully evaluated at the editorial level and by one of the previous peer reviewers. As you will see, reviewer #2 still has some major concerns regarding the work. We agree with those concerns; in particular, we note that in its current form, the manuscript is still difficult to follow. A more precise distinction of the population samples (both newly studied and collected from the literature) that were used as cases to test the hypotheses stated and the ones that were uses as reference populations, and a detailed description (in terms of geographic location and linguistic affiliation) of the whole dataset should be given in the beginning of results; and maybe also in beginning of the results’ sections, where the hypotheses being tested should also be described. There is also the lack of consistency in the naming of the populations throughout the paper, as mentioned by reviewer 2 – please be consistent on the naming of the population samples. Finally, the discussion section reads more as a summary of the results rather than a discussion.

Furthermore, we are concerned about overlap with the recent work from Kutanan et al. in MBE. Because that publication appeared during the revision of the current submission, we are willing to consider a revised manuscript for publication. However, for a revised manuscript to be successful, it will need to not only address concerns regarding reasoning and presentation, but also to explicitly acknowledge and comment on the work from Kutanan et al., noting which aspects of the current submission confirm, complement, and potentially extend the MBE paper.

Based on the reviews and our editorial discussion, we will not be able to accept this version of the manuscript, but we would be willing to review a much-revised version. We cannot, of course, promise publication at that time.

Should you decide to revise the manuscript for further consideration here, your revisions should address the specific points made by reviewer #2, the editorial concerns noted above, and specific editorial points listed below. We will also require a detailed list of your responses to the editorial and review comments and a description of the changes you have made in the manuscript.

If you decide to revise the manuscript for further consideration at PLOS Genetics, please aim to resubmit within the next 60 days, unless it will take extra time to address the concerns of the reviewers, in which case we would appreciate an expected resubmission date by email to plosgenetics@plos.org.

[LINK]

We are sorry that we cannot be more positive about your manuscript at this stage. Please do not hesitate to contact us if you have any concerns or questions.

Yours sincerely,

Sandra Beleza

Guest Editor

PLOS Genetics

Gregory Barsh

Editor-in-Chief

PLOS Genetics

Additional editorial comments:

Intro, Lines 89-93 – this is no longer true given Kutanan et al.Intro, lines 96 – this sentence sounds weird – please rephraseIntro, lines 111-113 – “Khmer from Thailand is a Northern Khmer speaking population, which is closely related to Khmer…” is also strange -> please rephrase to Cambodia Khmer ; also it is not really clear what you intended to say with Khmer being descendants of people from ancient Khmer states, since this would be what you expected; or is it that you want to say that the Khmer comprise an ancient group/ present in Thailand for a significant number of years (whose number should be described in the text)? There are other sentences like this when referring to other population groups that I ask the authors to rephrase.Results, lines 146-147 – change “is maximized” to “is enriched in”Results, line 153 – I think “biplot” instead of “scatterplot” is more specificResults, lines 180-182 – Sentences that start with a number, or that contain a number between 1-10, the numbers should be written down.Discussion, lines 324-325 – this is no longer true given Kutanan et al.Discussion, lines 517-518 – this is cryptic: what subtle differences or what similarities? How are these explained in terms of what is known about the history of the region?Table S1 – Could you please include the country and language-family of the populations used?Figure 5 – should read “Sgaw Karen”

Other comments regarding Analyses:

As noted by reviewer 2, there is a lack of evidence to support the splitting of the Thai groups.Please use the f3-statististics, qpAdmix and qpGraph to demonstrate new results that complement Kutanan et al.

Reviewer's Responses to Questions

**Comments to the Authors:**

Reviewer #2: I appreciate the authors making substantial revisions based on previous comments. While I think in the current version the authors at times still deviated from their main thesis of South Asian admixture in MSEA populations, the manuscript reads much better now.

My only major comment is still the practice of splitting of Thai into 3 groups. While the authors clarified that they decided to split Thai into 3 groups based on multiple analyses (PCA, ADMIXTURE, qpGraph, etc.), and I don’t deny that the seven Thai1 individuals may have a different admixture profile as the single Thai3 individual, the practice still appears arbitrary and unmerited. Given an admixed population, there are naturally variation in the admixture profile. That does not mean individuals at two ends of the spectrum are representatives of two populations. Imagine Latino individuals living in the United States. Sampling of Latino individuals from different parts of U.S., or even from the same city in U.S., will appear to have different proportion of ancestry. You might easily find some who carry African ancestry while others do not. That fact by itself does not mean there are multiple sub-populations within Latinos. By the authors’ own implicit admission, they have no other information on the Thai individuals (lines 455-456). Thus, there is no support from other sources (language, culture, self-identity, etc.) that would suggest it is appropriate to divide the Thai group into three. This is also based on very limited sampling (Thai2 and Thai3 have 2 and 1 individual, respectively). Such a contrived categorization ought to require a much larger study of the fine-scale structure of Thai, not based on a total of 10 individuals; taken to the extreme, this practice could be considered racist (discretizing a sample of individual based on their genetic profile). And ultimately, I see no additional scientific insight brought about by splitting Thai into three subgroups, other than the fact that “you can”.

Other comments:

1. The author should be more consistent when referring to populations in the manuscript. For example, multiple times they refer to “Indian” population (Line 147), but the Figure says “SAS”. There are also times they refer to Onge in the text, but NEG in the corresponding figure. In this case Onge and ESEA or South Asians are different levels of population labels.

2. The authors insisted on showing only the skeleton graph in Figure 6. I would prefer that the skeleton graph be moved to Supplemental Figure and instead show one or two graphs illustrating the main results instead. At the very least, a graph for each population examined in Figure 5 should be made in the supplement. Table S3 is very difficult to decipher and would help a lot with a graphical representation.

Other minor comments:

Line 43-44: this sentence should be tuned down a bit. There is evidence of gene flow, which “may have been” responsible for the spread of Indian culture in the region, but there’s no direct demonstration in this paper.

Line 71: reword “nearly substantial”

Line 134: should it be Eastern Southeast Asians for ESEA?

Line 146: it would be helpful if you can name the color of the ancestry component you’re referring to.

**Have all data underlying the figures and results presented in the manuscript been provided?**

Reviewer #2: **No: **Authors indicated data is on Reich's Lab website. I don't see them.

PLOS authors have the option to publish the peer review history of their article (what does this mean?). If published, this will include your full peer review and any attached files.

Reviewer #2: No

---

## [Decision Letter · Decision Letter 2]

12 Jan 2022

Dear Dr Changmai,

We are pleased to inform you that your manuscript entitled "Indian genetic heritage in Southeast Asian populations" has been editorially accepted for publication in PLOS Genetics. Congratulations!

Yours sincerely,

Sandra Beleza

Guest Editor

PLOS Genetics

Gregory Barsh

Editor-in-Chief

PLOS Genetics

Comments from the reviewers (if applicable):

Thank you for your amendments. I will now accept the manuscript

Reviewer's Responses to Questions

**Comments to the Authors:**

Reviewer #2: I appreciate the authors' response. I have no further comment of substance.

**Have all data underlying the figures and results presented in the manuscript been provided?**

Reviewer #2: Yes

PLOS authors have the option to publish the peer review history of their article (what does this mean?). If published, this will include your full peer review and any attached files.

Reviewer #2: No

**Data Deposition**

http://datadryad.org/submit?journalID=pgenetics&manu=PGENETICS-D-21-00062R2

**Press Queries**

---

## [Editor Report · Acceptance letter]

26 Jan 2022

PGENETICS-D-21-00062R2 

Indian genetic heritage in Southeast Asian populations 

Dear Dr Changmai, 

We are pleased to inform you that your manuscript entitled "Indian genetic heritage in Southeast Asian populations" has been formally accepted for publication in PLOS Genetics! Your manuscript is now with our production department and you will be notified of the publication date in due course.

With kind regards,

Zsofia Freund

PLOS Genetics

On behalf of:
